# Exploiting Causal Graph Priors with Posterior Sampling for Reinforcement Learning

**Mirco Mutti**[1][*] **Riccardo De Santi**[23] **Marcello Restelli**[4] **Alexander Marx**[2][†] **Giorgia Ramponi**[3][†]
[1]Technion, [2]ETH Zurich, [3]ETH AI Center, [4]Politecnico di Milano
`mirco.m@technion.ac.il`, `rdesanti@ethz.ch`, `marcello.restelli@polimi.it`
`alexander.marx@ai.ethz.ch`, `giorgia.ramponi@ai.ethz.ch`

## Abstract

Posterior sampling allows exploitation of prior knowledge on the environment's transition dynamics to improve the sample efficiency of reinforcement learning. The prior is typically specified as a class of parametric distributions, the design of which can be cumbersome in practice, often resulting in the choice of uninformative priors. In this work, we propose a novel posterior sampling approach in which the prior is given as a (partial) causal graph over the environment's variables. The latter is often more natural to design, such as listing known causal dependencies between biometric features in a medical treatment study. Specifically, we propose a hierarchical Bayesian procedure, called C-PSRL, simultaneously learning the full causal graph at the higher level and the parameters of the resulting factored dynamics at the lower level. We provide an analysis of the Bayesian regret of C-PSRL that explicitly connects the regret rate with the degree of prior knowledge. Our numerical evaluation conducted in illustrative domains confirms that C-PSRL strongly improves the efficiency of posterior sampling with an uninformative prior while performing close to posterior sampling with the full causal graph.

## 1 Introduction

Posterior sampling (Thompson, 1933), a.k.a. Thompson sampling, is a powerful alternative to classic optimistic methods for Reinforcement Learning (RL, Sutton & Barto, 2018) as it guarantees outstanding sample efficiency (Osband et al., 2013) through an explicit model of the epistemic uncertainty that allows exploiting prior knowledge over the environment's dynamics. Specifically, Posterior Sampling for Reinforcement Learning (PSRL, Strens, 2000; Osband et al., 2013) implements a Bayesian procedure in which, at every episode $k$, (1) a model of the environment's dynamics is sampled from a parametric prior distribution $P_k$, (2) an optimal policy $\pi_k$ is computed (e.g., through value iteration (Bellman, 1957)) according to the sampled model, (3) a posterior update is performed on the prior parameters to incorporate in $P_{k+1}$ the evidence collected by running $\pi_k$ in the true environment. Under the assumption that the true environment's dynamics are sampled with positive probability from the prior $P_0$, the latter procedure is provably efficient as it showcases a Bayesian regret that scales with $O(\sqrt{K})$ being $K$ the total number of episodes (Osband & Van Roy, 2017).

Although posterior sampling has been also praised for its empirical prowess (Chapelle & Li, 2011), specifying the prior through a class of parametric distributions, a crucial requirement of PSRL, can be cumbersome in practice. Let us take a Dynamic Treatment Regime (DTR, Murphy, 2003) as an illustrative application. Here, we aim to overcome a patient's disease by choosing, at each stage, a treatment based on the patient's evolving conditions and previously administered treatments. The goal is to identify the best treatment for the specific patient quickly. Medicine provides plenty of prior knowledge to help solve the DTR problem. However, it is not easy to translate this knowledge into a parametric prior distribution that is general enough to include the model of any patient while being sufficiently narrow to foster efficiency. Instead, it is remarkably easy to list some known causal relationships between patient's state variables, such as heart rate and blood pressure, or diabetes and glucose level. Those causal edges might come from experts' knowledge (e.g., physicians) or previous clinical studies. A prior in the form of a causal graph is more natural to specify for practitioners,

---

[*]Work done while the author was at Politecnico di Milano.   [†]Joint senior-authorship.

who might be unaware of the intricacies of Bayesian statistics. Posterior sampling does not currently support the specification of the prior through a causal graph, which limits its applicability.

This paper proposes a novel posterior sampling methodology that can exploit a prior specified through a partial causal graph over the environment's variables. Notably, a complete causal graph allows for a factorization of the environment's dynamics, which can be then expressed as a Factored Markov Decision Process (FMDP, Boutilier et al., 2000). PSRL can be applied to FMDPs, as demonstrated by previous work (Osband & Van Roy, 2014), where the authors assume to know the complete causal graph. However, this assumption is often unreasonable in practical applications.[1]

Instead, we assume to have partial knowledge of the causal graph, which leads to considering a set of plausible FMDPs. Taking inspiration from (Hong et al., 2020; 2022b;a; Kveton et al., 2021), we design a hierarchical Bayesian procedure, called **Causal PSRL** (C-PSRL), extending PSRL to the setting where the true model lies within a set of FMDPs (induced by the causal graph prior). At each episode, C-PSRL first samples a factorization consistent with the causal graph prior. Then, it samples the model of the FMDP from a lower-level prior that is conditioned on the sampled factorization. After that, the algorithm proceeds similarly to PSRL on the sampled FMDP.

Having introduced C-PSRL, we study the Bayesian regret it induces on the footsteps of previous analyses for PSRL in FMDPs (Osband & Van Roy, 2014) and hierarchical posterior sampling (Hong et al., 2022b). Our analysis shows that C-PSRL takes the best of both worlds by avoiding a direct dependence on the number of states in the regret (as in FMDPs) and without requiring a full causal graph prior (as in hierarchical posterior sampling). Moreover, we can analytically capture the dependency of the Bayesian regret on the number of causal edges known a priori and encoded in the (partial) causal graph prior. Finally, we empirically validate C-PSRL against two relevant baselines: PSRL with an uninformative prior, i.e., that does not model potential factorizations in the dynamics, and PSRL equipped with the full knowledge of the causal graph (an oracle prior). We carry out the comparison in simple yet illustrative domains, which show that exploiting a causal graph prior improves efficiency over uninformative priors while being only slightly inferior to the oracle prior.

In summary, the main contributions of this paper include the following:
- A novel problem formulation that links PSRL with a prior expressed as a partial causal graph to the problem of learning an FMDP with unknown factorization (Section 2);
- A methodology (C-PSRL) that extends PSRL to exploit a partial causal graph prior (Section 3);
- The analysis of the Bayesian regret of C-PSRL, which is $\tilde{O}(\sqrt{K/2^\eta})^2$ where $K$ is the total number of episodes and $\eta$ is the degree of prior knowledge (Section 4);
- An ancillary result on causal discovery that shows how a (sparse) super-graph of the true causal graph can be extracted from a run of C-PSRL as a byproduct (Section 5);
- An experimental evaluation of the performance of C-PSRL against PSRL with uninformative or oracle priors in illustrative domains (Section 6).

Finally, the aim of this work is to enable the use of posterior sampling for RL in relevant applications through a causal perspective on prior specification. We believe this contribution can help to close the gap between PSRL research and actual adoption of PSRL methods in real-world problems.

## 2 PROBLEM FORMULATION

In this section, we first provide preliminary background on graphical causal models (Section 2.1) and Markov decision processes (Section 2.2). Then, we explain how a causal graph on the variables of a Markov decision process induces a factorization of its dynamics (Section 2.3). Finally, we formalize the reinforcement learning problem in the presence of a causal graph prior (Section 2.4).

**Notation.**  With few exceptions, we will denote a set or space as $\mathcal{A}$, their elements as $a \in \mathcal{A}$, constants or random variables with $A$, and functions as $f$. We denote $\Delta(\mathcal{A})$ the probability simplex over $\mathcal{A}$, and $[A]$ the set of integers $\{1, \ldots, A\}$. For a $d$-dimensional vector $x$, we define the *scope operator* $x[\mathcal{I}] := \bigotimes_{i \in \mathcal{I}} x_i$ for any set $\mathcal{I} \subseteq [d]$. When $\mathcal{I} = \{i\}$ is a singleton, we use $x[i]$ as a shortcut for $x[\{i\}]$. A recap of the notation, which is admittedly involved, can be found in Appendix A.

---

[1] DTR is an example, where several causal relations affecting patient's conditions remain a mystery.

[2] We report regret rates with the common "Big-O" notation, in which $\tilde{O}$ hides logarithmic factors. Note that the rate here is simplified to highlight the most relevant factors. A complete rate can be found in Theorem 4.1.

## 2.1 CAUSAL GRAPHS

Let $\mathcal{X} = \{X_j\}_{j=1}^{d_X}$ and $\mathcal{Y} = \{Y_j\}_{j=1}^{d_Y}$ be sets of random variables taking values $x_j, y_j \in [N]$ respectively, and let $p : \mathcal{X} \to \Delta(\mathcal{Y})$ a strictly positive probability density. Further, let $\mathcal{G} = (\mathcal{X}, \mathcal{Y}, z)$ be a bipartite Directed Acyclic Graph (DAG), or *bigraph*, having left variables $\mathcal{X}$, right variables $\mathcal{Y}$, and a set of edges $z \subseteq \mathcal{X} \times \mathcal{Y}$. We denote as $z_j$ the parents of the variable $Y_j \in \mathcal{Y}$, such as $z_j = \{i \in [d_X] \mid (X_i, Y_j) \in z\}$ and $z = \bigcup_{j \in [d_Y]} \bigcup_{i \in z_j} \{(X_i, Y_j)\}$. We say that $\mathcal{G}$ is $Z$-sparse if $\max_{j \in [d_Y]} |z_j| \le Z \le d_X$, and we call $Z$ the *degree of sparseness* of $\mathcal{G}$.

The tuple $(p, \mathcal{G})$ is called a *graphical causal model* (Pearl, 2009) if $p$ fulfills the Markov factorization property with respect to $\mathcal{G}$, that is $p(\mathcal{X}, \mathcal{Y}) = p(\mathcal{X})p(\mathcal{Y}|\mathcal{X}) = p(\mathcal{X}) \prod_{j \in [d_Y]} p_j(y[j]|x[z_j])$ and all interventional distributions are well defined.[3] Note that the causal model that we consider in this paper does not admit *confounding*. Further, we can exclude "vertical" edges in $\mathcal{Y} \times \mathcal{Y}$ and directed edges $\mathcal{Y} \times \mathcal{X}$. Finally, we call *causal graph* the component $\mathcal{G}$ of a graphical causal model.

## 2.2 MARKOV DECISION PROCESSES

A finite episodic Markov Decision Process (MDP, Puterman, 2014) is defined throug the tuple $\mathcal{M} := (\mathcal{S}, \mathcal{A}, p, r, \mu, H)$, where $\mathcal{S}$ is a state space of size $S$, $\mathcal{A}$ is an action space of size $A$, $p : \mathcal{S} \times \mathcal{A} \to \Delta(\mathcal{S})$ is a Markovian transition model such that $p(s'|s, a)$ denotes the conditional probability of the next state $s'$ given the state $s$ and action $a$, $r : \mathcal{S} \times \mathcal{A} \to \Delta([0, 1])$ is a reward function such that the reward collected performing action $a$ in state $s$ is distributed as $r(s, a)$ with mean $R(s, a) = \mathbb{E}[r(s, a)]$, $\mu \in \Delta(\mathcal{S})$ is the initial state distribution, $H < \infty$ is the episode horizon.

An agent interacts with the MDP as follows. First, the initial state is drawn $s_1 \sim \mu$. For each step $h < H$, the agent selects an action $a_h \in \mathcal{A}$. Then, they collect a reward $r_h \sim r(s_h, a_h)$ while the state transitions to $s_{h+1} \sim p(\cdot|s_h, a_h)$. The episode ends when $s_H$ is reached.

The strategy from which the agent selects an action at each step is defined through a non-stationary, stochastic *policy* $\pi = \{\pi_h\}_{h \in [H]} \in \Pi$, where each $\pi_h : \mathcal{S} \to \Delta(\mathcal{A})$ is a function such that $\pi_h(a|s)$ denotes the conditional probability of selecting action $a$ in state $s$ at step $h$, and $\Pi$ is the policy space. A policy $\pi \in \Pi$ can be evaluated through its value function $V_h^\pi : \mathcal{S} \to [0, H]$, which is the expected sum of rewards collected under $\pi$ starting from state $s$ at step $h$, i.e.,

$$V_h^\pi(s) := \mathbb{E}_\pi \left[ \sum_{h'=h}^{H} R(s_{h'}, a_{h'}) \Big| s_h = s \right], \qquad \forall s \in \mathcal{S},\ h \in [H].$$

We further define the value function of $\pi$ in the MDP $\mathcal{M}$ under $\mu$ as $V_\mathcal{M}(\pi) := \mathbb{E}_{s \sim \mu}[V_1^\pi(s)]$.

## 2.3 CAUSAL STRUCTURE INDUCES FACTORIZATION

In the previous section, we formulated the MDP in a tabular representation, where each state (action) is identified by a unique symbol $s \in \mathcal{S}$ ($a \in \mathcal{A}$). However, in relevant real-world applications, the states and actions may be represented through a finite number of features, say $d_S$ and $d_A$ features respectively. The DTR problem is an example, where state features can be, e.g., blood pressure and glucose level, action features can be indicators on whether a particular medication is administered.

Let those state and action features be modeled by random variables in the interaction between an agent and the MDP, we can consider additional structure in the process by considering the causal graph of its variables, such that the value of a variable only depends on the values of its causal parents. Looking back to DTR, we might know that the value of the blood pressure at step $h + 1$ only depends on its value at step $h$ and whether a particular medication has been administered.

Formally, we can show that combining an MDP $\mathcal{M} = (\mathcal{S}, \mathcal{A}, p, r, \mu, H)$ with a causal graph over its variables, which we denote as $\mathcal{G}_\mathcal{M} = (\mathcal{X}, \mathcal{Y}, z)$, gives a factored MDP (Boutilier et al., 2000)

$$\mathcal{F} = (\{\mathcal{X}_j\}_{j=1}^{d_X}, \{\mathcal{Y}_j, z_j, p_j, r_j\}_{j=1}^{d_Y}, \mu, H, Z, N),$$

---

[3]Both assumptions come naturally with factored MDPs (see Section 2.3). Informally, an intervention on a node $Y_j \in \mathcal{Y}$ (or in $\mathcal{X}$) assigns $y[j]$ to a constant $c$, i.e., $p_j(y[j] = c) = 1$. All mechanisms $p_i$, where $i$ is not intervened on, remain invariant. For more details on the causal notation we refer to (Pearl, 2009, Chapter 1).

Figure 1: **(Left)** Illustrative causal graph prior $\mathcal{G}_0$ with $d_X = 4, d_Y = 2$ features, degree of sparseness $Z = 3$. The hidden true graph $\mathcal{G}_{\mathcal{F}_*}$ includes all the edges in $\mathcal{G}_0$ plus the red-dashed edge $(3, 1)$. **(Right)** Visualization of $\mathcal{Z}$, the set of factorizations consistent with $\mathcal{G}_0$, which is the support of the hyper-prior $P_0$. The factorization $z_*$ of the true FMDP $\mathcal{F}_*$ is highlighted in red.

where $\mathcal{X} = \mathcal{S} \times \mathcal{A} = \mathcal{X}_1 \times \ldots \times \mathcal{X}_{d_X}$ is a factored state-action space with $d_X = d_S + d_A$ discrete variables, $\mathcal{Y} = \mathcal{S} = \mathcal{Y}_1 \times \ldots \times \mathcal{Y}_{d_Y}$ is a factored state space with $d_Y = d_S$ variables, and $z_j$ are the causal parents of each state variable, which are obtained from the edges $z$ of $\mathcal{G}_{\mathcal{M}}$. Then, $p$ is a *factored* transition model specified as $p(y|x) = \prod_{j=1}^{d_Y} p_j(y[j] \mid x[z_j]), \forall y \in \mathcal{Y}, x \in \mathcal{X}$, and $r$ is a *factored* reward function $r(x) = \sum_{j=1}^{d_Y} r(x[z_j])$, with mean $R(x) = \sum_{j=1}^{d_Y} R(x[z_j]), \forall x \in \mathcal{X}$. Finally, $\mu \in \Delta(\mathcal{Y})$ and $H$ are the initial state distribution and episode horizon as specified in $\mathcal{M}$, $Z$ is the degree of sparseness of $\mathcal{G}_{\mathcal{M}}$, $N$ is a constant such that all the variables are supported in $[N]$.

The interaction between an agent and the FMDP can be described exactly as we did in Section 2.2 for a tabular MDP, and the policies with their corresponding value functions are analogously defined. With the latter formalization of the FMDP induced by a causal graph, we now have all the components to introduce our learning problem in the next section.

## 2.4 REINFORCEMENT LEARNING WITH PARTIAL CAUSAL GRAPH PRIORS

In the previous section, we show how the prior knowledge of a causal graph over the MDP variables can be exploited to obtain an FMDP representation of the problem, which is well-known to allow for more efficient reinforcement learning thanks to the factorization of the transition model and reward function (Osband & Van Roy, 2014; Xu & Tewari, 2020; Tian et al., 2020; Chen et al., 2020; Talebi et al., 2021; Rosenberg & Mansour, 2021). However, in several applications is unreasonable to assume prior knowledge of the full causal graph, and causal identification is costly in general (Gillispie & Perlman, 2001; Shah & Peters, 2020). Nonetheless, *some* prior knowledge of the causal graph, i.e., a portion of the edges, may be easily available. For instance, in a DTR problem some edges of the causal graph on patient's variables are commonly known, whereas several others are elusive.

In this paper, we study the reinforcement learning problem when a partial causal graph prior $\mathcal{G}_0 \subseteq \mathcal{G}_{\mathcal{M}}$ on the MDP $\mathcal{M}$ is available.[4] We formulate the learning problem in a Bayesian sense, in which the instance $\mathcal{F}_*$ is sampled from a prior distribution $P_{\mathcal{G}_0}$ *consistent* with the causal graph prior $\mathcal{G}_0$.[5] In Figure 1 (left), we illustrate both the causal graph prior $\mathcal{G}_0$ and the (hidden) true graph $\mathcal{G}_{\mathcal{F}_*}$ of the true instance $\mathcal{F}_*$. Analogously to previous works on Bayesian RL formulations, e.g., (Osband et al., 2013), we evaluate the performance of a learning algorithm in terms of its induced Bayesian regret.

**Definition 1** (Bayesian Regret). *Let $\mathfrak{A}$ a learning algorithm and let $P_{\mathcal{G}_0}$ a prior distribution on FMDPs consistent with the partial causal graph prior $\mathcal{G}_0$. The $K$-episodes Bayesian regret of $\mathfrak{A}$ is*

$$\mathcal{BR}(K) := \mathop{\mathbb{E}}_{\mathcal{F}_* \sim P_{\mathcal{G}_0}} \left[ \sum_{k=1}^{K} V_*(\pi_*) - V_*(\pi_k) \right],$$

*where $V_*(\pi) = V_{\mathcal{F}_*}(\pi)$ is the value of the policy $\pi$ in $\mathcal{F}_*$ under $\mu$, $\pi_* \in \arg\max_{\pi \in \Pi} V_*(\pi)$ is the optimal policy in $\mathcal{F}_*$, and $\pi_k$ is the policy played by algorithm $\mathfrak{A}$ at step $k \in [K]$.*

The Bayesian regret allows to evaluate a learning algorithm on average over multiple instances. This is particularly suitable in some domains, such as DTR, in which it is crucial to achieve a good performance of the treatment policy on different patients. In the next section, we introduce an algorithm that achieves a Bayesian regret rate that is sublinear in the number of episodes $K$.

---

[4]For two bigraphs $\mathcal{G}_\star = (\mathcal{X}, \mathcal{Y}, z_\star)$ and $\mathcal{G}_\bullet = (\mathcal{X}, \mathcal{Y}, z_\bullet)$, we let $\mathcal{G}_\star \subseteq \mathcal{G}_\bullet$ if $z_\star \subseteq z_\bullet$.

[5]We will specify in the next Section 3 how the prior $P_{\mathcal{G}_0}$ can be constructed.

## 3 CAUSAL PSRL

To face the learning problem described in the previous section, we cannot naïvely apply the PSRL algorithm for FMDPs (Osband & Van Roy, 2014), since we cannot access the factorization $z_*$ of the true instance $\mathcal{F}_*$, but only a causal graph prior $\mathcal{G}_0 = (\mathcal{X}, \mathcal{Y}, z_0)$ such that $z_0 \subseteq z_*$. Moreover, $z_*$ is always *latent* in the interaction process, in which we can only observe state-action-reward realizations from $\mathcal{F}_*$. The latter can be consistent with several factorizations of the transition dynamics of $\mathcal{F}_*$, which means we can neither extract $z_*$ directly from data. This is the common setting of *hierarchical Bayesian methods* (Hong et al., 2020; 2022a;b; Kveton et al., 2021), where a latent state is sampled from a latent hypothesis space on top of the hierarchy, which then conditions the sampling of the observed state down the hierarchy. In our setting, we can see the latent hypothesis space as the space of all the possible factorizations that are consistent with $\mathcal{G}_0$, whereas the observed states are the model parameters of the FMDP, from which we observe realizations. The algorithm that we propose, **C**ausal **PSRL** (C-PSRL), builds on this intuition to implement a principled hierarchical posterior sampling procedure to minimize the Bayesian regret exploiting the causal graph prior.

---

**Algorithm 1** Causal PSRL (C-PSRL)

---

1: **input**: causal graph prior $\mathcal{G}_0 \subseteq \mathcal{G}_{\mathcal{F}_*}$, degree of sparseness $Z$
2: Compute the set of consistent factorizations
$$\mathcal{Z} = \mathcal{Z}_1 \times \ldots \times \mathcal{Z}_{d_Y} = \left\{ z = \{z_j\}_{j \in [d_Y]} \; \middle| \; |z_j| < Z \text{ and } z_{0,j} \subseteq z_j \quad \forall j \in [d_Y] \right\}$$
3: Build the hyper-prior $P_0$ and the prior $P_0(\cdot|z)$ for each $z \in \mathcal{Z}$
4: **for** episode $k = 0, 1, \ldots, K-1$ **do**
5:     Sample $z \sim P_k$ and $p \sim P_k(\cdot|z)$ to build the FMDP $\mathcal{F}_k$
6:     Compute the policy $\pi_k \leftarrow \arg\max_{\pi \in \Pi} V_{\mathcal{F}_k}(\pi)$ collect an episode with $\pi_k$ in $\mathcal{F}_*$
7:     Compute the posteriors $P_{k+1}$ and $P_{k+1}(\cdot|z)$ with the collected data
8: **end for**

---

First, C-PSRL computes the set $\mathcal{Z}$, illustrated in Figure 1 (right), of the factorizations consistent with $\mathcal{G}_0$, i.e., which are both $Z$-sparse and include all of the edges in $z_0$ (line 2). Then, it specifies a parametric distribution $P_0$, called *hyper-prior*, over the latent hypothesis space $\mathcal{Z}$, and, for each $z \in \mathcal{Z}$, a further parametric distribution $P_0(\cdot|z)$, which is a *prior* on the model parameters, i.e., transition probabilities, conditioned on the latent state $z$ (line 3). The former represents the agent's belief over the factorization of the true instance $\mathcal{F}_*$, the latter on the factored transition model $p_*$.[6]

Having translated the causal graph prior $\mathcal{G}_0$ into proper parametric prior distributions, C-PSRL executes a hierarchical posterior sampling procedure (lines 4-8). For each episode $k$, the algorithm sample a factorization $z$ from the current hyper-prior $P_k$, and a transition model $p$ from the prior $P_k(\cdot|z)$, such that $p$ is factored according to $z$ (line 5). With these two, it builds the FMDP $\mathcal{F}_k$ (line 5), for which it computes the optimal policy $\pi_k$ solving the corresponding planning problem, which is deployed on the true instance $\mathcal{F}_*$ for one episode (line 6). Finally, the evidence collected in $\mathcal{F}_*$ serves to compute the closed-form posterior updates of the prior and hyper-prior (line 7).

As we shall see, Algorithm 1 has compelling statistical properties, a regret sublinear in $K$ (Section 4) with a notion of causal discovery (Section 5), and promising empirical performance (Section 6).

**Recipe.** Three key ingredients concur to make the algorithm successful. First, C-PSRL links RL of an FMDP $\mathcal{F}_*$ with unknown factorization to a hierarchical Bayesian learning, in which the factorization acts as a latent state on top of the hierarchy, and the transition probabilities are the observed state down the hierarchy. Secondly, C-PSRL exploits the causal graph prior $\mathcal{G}_0$ to reduce the size of the latent hypothesis space $\mathcal{Z}$, which is super-exponential in $d_X, d_Y$ in general (Robinson, 1973). Finally, C-PSRL harnesses the specific causal structure of the problem to get a factorization $z$ (line 5) through independent sampling of the parents $z_j \in \mathcal{Z}_j$ for each $Y_j$, which significantly reduces the number of hyper-prior parameters. Crucially, this can be done as we do not admit "vertical" edges in $\mathcal{Y}$ and edges from $\mathcal{Y}$ to $\mathcal{X}$, such that parents' assignment cannot lead to a cycle.

**Degree of sparseness.** C-PSRL takes as input (line 1) the degree of sparseness $Z$ of the true FMDP $\mathcal{F}_*$, which might be unknown in practice. In that case, $Z$ can be seen as an hyper-parameter of the algorithm, which can be either implied through domain expertise or tuned independently.

---

[6]A description of parametric distributions $P_0$ and $P_0(\cdot|z)$ and their posterior updates is in Appendix B.

**Planning in FMDPs.** C-PSRL requires exact planning in a FMDP (line 6), which is intractable in general (Mundhenk et al., 2000; Lusena et al., 2001). While we do not address computational issues in this paper, we note that efficient approximation schemes have been developed (Guestrin et al., 2003). Moreover, under linear realizability assumptions for the transition model or value functions, exact planning methods exist (Yang & Wang, 2019; Jin et al., 2020b; Deng et al., 2022).

## 4 REGRET ANALYSIS OF C-PSRL

In this section, we study the Bayesian regret induced by C-PSRL with a $Z$-sparse causal graph prior $\mathcal{G}_0 = (\mathcal{X}, \mathcal{Y}, z_0)$. First, we define the *degree of prior knowledge* $\eta \leq \min_{j \in [d_Y]} |z_{0,j}|$, which is a lower bound on the number of causal parents revealed by the prior $\mathcal{G}_0$ for each state variable $Y_j$. We then provide an upper bound on the Bayesian regret of C-PSRL, which we discuss in Section 4.1.

**Theorem 4.1.** *Let $\mathcal{G}_0$ be a causal graph prior with degree of sparseness $Z$ and degree of prior knowledge $\eta$. The $K$-episodes Bayesian regret incurred by* C-PSRL *is*

$$\mathcal{BR}(K) = \tilde{O}\left(\left(H^{5/2} N^{1+Z/2} d_Y + \sqrt{H 2^{d_X - \eta}}\right) \sqrt{K}\right).$$

While we defer the proof of the result to Appendix E, we report a sketch of its main steps below.

**Step 1.** The first step of our proof bridges the previous analyses of a hierarchical version of PSRL, which is reported in (Hong et al., 2022b), with the one of PSRL for factored MDPs (Osband & Van Roy, 2014). In short, we can decompose the Bayesian regret (see Definition 1) as

$$\mathcal{BR}(K) = \mathbb{E}\left[\sum_{k=1}^{K} \mathbb{E}_k \left[V_*(\pi_*) - \overline{V}_k(\pi_*, Z_*)\right]\right] + \mathbb{E}\left[\sum_{k=1}^{K} \mathbb{E}_k \left[\overline{V}_k(\pi_k, Z_k) - V_*(\pi_k)\right]\right]$$

where $\mathbb{E}_k[\cdot]$ is the conditional expectation given the evidence collected until episode $k$, and $\overline{V}_k(\pi, z) = \mathbb{E}_{\mathcal{F} \sim P_k(\cdot|z)}[V_{\mathcal{F}}(\pi)]$ is the value function of $\pi$ on average over the posterior $P_k(\cdot|z)$. Informally, the first term captures the regret due to the concentration of the *posterior* $P_k(\cdot|z_*)$ around the true transition model $p_*$ having fixed the true factorization $z_*$. Instead, the second term captures the regret due to the concentration of the *hyper-posterior* $P_k$ around the true factorization $z_*$. Through a non-trivial adaptation of the analysis in (Hong et al., 2022b) to the FMDP setting, we can bound each term separately to obtain $\tilde{O}((H^{5/2} N^{1+Z/2} d_Y + \sqrt{H|\mathcal{Z}|})\sqrt{K})$.

**Step 2.** The upper bound of the previous step is close to the final result up to a factor $\sqrt{|\mathcal{Z}|}$ related the size of the latent hypothesis space. Since C-PSRL performs local sampling from the product space $\mathcal{Z} = \mathcal{Z}_1 \times \ldots \times \mathcal{Z}_{d_Y}$, by combining independent samples $z_j \in \mathcal{Z}_j$ for each variable $Y_j$ as we briefly explained in Section 3, we can refine the dependence in $|\mathcal{Z}|$ to $\max_{j \in [d_Y]} |\mathcal{Z}_j| \leq |\mathcal{Z}|$.

**Step 3.** Finally, to obtain the final rate reported in Theorem 4.1, we have to capture the dependency in the degree of prior knowledge $\eta$ in the Bayesian regret by upper bounding $\max_{j \in [d_Y]} |\mathcal{Z}_j|$ as

$$\max_{j \in [d_Y]} |\mathcal{Z}_j| = \sum_{i=0}^{Z-\eta} \binom{d_X - \eta}{i} \leq 2^{d_X - \eta}.$$

### 4.1 DISCUSSION OF THE BAYESIAN REGRET

The regret bound in Theorem 4.1 contains two terms, which informally capture the regret to learn the transition model having the true factorization (left), and to learn the true factorization (right).

The first term is typical in previous analyses of vanilla posterior sampling. Especially, the best known rate for the MDP setting is $\tilde{O}(H\sqrt{SAK})$ (Osband & Van Roy, 2017). In a FMDP setting with known factorization, the direct dependencies with the size $S, A$ of the state and action spaces can be refined to obtain $\tilde{O}(H d_Y^{3/2} N^{Z/2} \sqrt{K})$ (Osband & Van Roy, 2014). Our rate includes additional factors of $H$ and $N$, but a better dependency on the number of state features $d_Y$.

The second term of the regret rate is instead unique to hierarchical Bayesian settings, which include an additional source of randomization in the sampling of the latent state from the hyper-prior. In Theorem 4.1, we are able to express this term in the degree of prior knowledge $\eta$, resulting in a rate

$\tilde{O}(\sqrt{K/2^\eta})$. The latter naturally demonstrates that a richer causal graph prior $\mathcal{G}_0$ will benefit the efficiency of PSRL, bringing the regret rate closer to the one for an FMDP with known factorization.

We believe that the rate in Theorem 4.1 is shedding light on how prior causal knowledge, here expressed through a partial causal graph, impacts on the efficiency of posterior sampling for RL.

## 5  C-PSRL EMBEDS A NOTION OF CAUSAL DISCOVERY

In this section, we provide an ancillary result that links Bayesian regret minimization with C-PSRL to a notion of causal discovery, which we call *weak causal discovery*. Especially, we show that we can extract a $Z$-sparse super-graph of the causal graph $\mathcal{G}_{\mathcal{F}_*}$ of the true instance $\mathcal{F}_*$ as a byproduct.

A run of C-PSRL produces a sequence $\{\pi_k\}_{k=0}^{K-1}$ of optimal policies for the FMDPs $\{\mathcal{F}_k\}_{k=0}^{K-1}$ drawn from the posteriors. Every FMDP $\mathcal{F}_k$ is linked to a corresponding graph (or factorization) $\mathcal{G}_{\mathcal{F}_k} = (\mathcal{X}, \mathcal{Y}, z_k)$, where $z_k \sim P_k$ is sampled from the hyper-posterior. Note that the algorithm does not enforce any causal meaning to the edges $z_k$ of $\mathcal{G}_{\mathcal{F}_k}$. Nonetheless, we aim to show that we can extract a $Z$-sparse super-graph of $\mathcal{G}_{\mathcal{F}_*}$ from the sequence $\{\mathcal{F}_k\}_{k=0}^{K-1}$ with high probability.

First, we need to assume that any misspecification in $\mathcal{G}_{\mathcal{F}_k}$ negatively affects the value function of $\pi_k$. Thus, we extend the traditional notion of causal minimality (Spirtes et al., 2000) to value functions.

**Definition 2** ($\epsilon$-Value Minimality). *An FMDP $\mathcal{F}$ fulfills $\epsilon$-value minimality, if for any FMDP $\mathcal{F}'$ encoding a proper subgraph of $\mathcal{G}_\mathcal{F}$, i.e., $\mathcal{G}_{\mathcal{F}'} \subset \mathcal{G}_\mathcal{F}$, it holds that $V_\mathcal{F}^* > V_{\mathcal{F}'}^* + \epsilon$, where $V_\mathcal{F}^*, V_{\mathcal{F}'}^*$ are the value functions of the optimal policies in $\mathcal{F}, \mathcal{F}'$ respectively.*

Then, as a corollary of Theorem 4.1, we can prove the following result.

**Corollary 5.1** (Weak Causal Discovery). *Let $\mathcal{F}_*$ be an FMDP in which the transition model $p_*$ fulfills the causal minimality assumption with respect to $\mathcal{G}_{\mathcal{F}_*}$, and let $\mathcal{F}_*$ fulfill $\epsilon$-value minimality. Then, $\mathcal{G}_{\mathcal{F}_*} \subseteq \mathcal{G}_{\mathcal{F}_K}$ holds with high probability, where $\mathcal{G}_{\mathcal{F}_K}$ is a $Z$-sparse graph randomly selected within the sequence $\{\mathcal{G}_{\mathcal{F}_k}\}_{k=0}^{K-1}$ produced by C-PSRL over $K = \tilde{O}(H^5 d_Y^2 2^{d_X - \eta}/\epsilon^2)$ episodes.*

The latter result shows that C-PSRL discovers the causal relationships between the FMDP variables, but cannot easily prune the non-causal edges, making $\mathcal{G}_{\mathcal{F}_K}$ a super-graph of $\mathcal{G}_{\mathcal{F}_*}$. In Appendix D, we report a detailed derivation of the previous result. Interestingly, Corollary 5.1 suggests a direct link between regret minimization in a FMDP with unknown factorization and a (weak) notion of causal discovery, which might be further explored in future works.

## 6  EXPERIMENTS

In this section, we provide experiments to both support the design of C-PSRL (Section 3) and validate its regret rate (Section 4). We consider two simple yet illustrative domains. The first, which we call *Random FMDP*, benchmarks the performance of C-PSRL on randomly generated FMDP instances, a setting akin to the Bayesian learning problem (see Section 2.4) that we considered in previous sections. The latter is a traditional *Taxi* environment (Dieterich, 2000), which is naturally factored and hints at a potential application. In those domains, we compare C-PSRL against two natural baselines: PSRL for tabular MDPs (Strens, 2000) and Factored PSRL (F-PSRL), which extends PSRL to factored MDP settings (Osband & Van Roy, 2014). Note that F-PSRL is equivalent to an instance of C-PSRL that receives the true causal graph prior as input, i.e., has an oracle prior.

**Random FMDPs.**  An FMDP (relevant parameters are reported in the caption of Figure 2) is sampled uniformly from the prior specified through a random causal graph, which is $Z$-sparse with at least two edges for every state variable ($\eta = 2$). Then, the regret is minimized by running PSRL, F-PSRL, and C-PSRL ($\eta = 2$) for 500 episodes. Figure 2a shows that C-PSRL achieves a regret that is significantly smaller than PSRL, thus outperforming the baseline with an uninformative prior, while being surprisingly close to F-PSRL, having the oracle prior. Indeed, C-PSRL resulted efficient in estimating the transition model of the sampled FMDP, as we can see from Figure 2b, which reports the $\ell_1$ distance between the true model $p_*$ and the $p_k$ sampled by the algorithm at episode $k$.

**Taxi.**  For the Taxi domain, we use the common Gym implementation (Brockman et al., 2016). In this environment, a taxi driver needs to pick up a passenger at a specific location, and then it has to

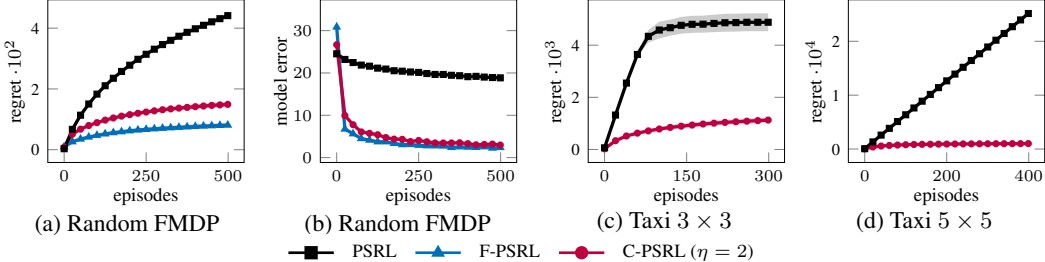

Figure 2: **(a,b)** Regret and model error as a function of the episodes in the Random FMDP domain with $d_X = 9, d_Y = 6, Z = 5, N = 2, H = 100$. **(c,d)** Regret as a function of the episodes in Taxi $3 \times 3$ with $d_X = 5, d_Y = 4, Z = 5, N = [3, 3, 2, 1, 6], H = 10$, Taxi $5 \times 5$ with $d_X = 5, d_Y = 4, Z = 5, N = [5, 5, 2, 1, 6], H = 15$. The plots report the mean and 95% c.i. over 20 runs.

bring the passenger to their destination. The environment is represented as a grid, with some special cells identifying the passenger location and destination. As reported in Simão & Spaan (2019), this domain is inherently factored since the state space is represented by four independent features: The position of the taxi (row and column), the passenger's location and whether they are on the taxi, and the destination. We perform the experiment on two grids with varying size ($3 \times 3$ and $5 \times 5$ respectively), for which we report the relevant parameters in Figure 2. Here we compare the proposed algorithm C-PSRL ($\eta = 2$) with PSRL, while F-PSRL is omitted as the knowledge of the oracle prior is not available. Both algorithms converge to a good policy eventually in the smaller grid (see the regret in Figure 2c). Instead, when the size of the grid increases, PSRL is still suffering a linear regret after 400 episodes, whereas C-PSRL succeeds in finding a good policy efficiently (see Figure 2d). Notably, this domain resembles the problem of learning optimal routing in a taxi service, and our results show that exploiting common knowledge (such as that the location of the taxi and passenger's destination) in the form of a causal graph prior can be a game changer.

## 7 RELATED WORK

We revise here the most relevant related work in posterior sampling, factored MDPs, and causal RL.

**Posterior sampling.** Thompson sampling (Thompson, 1933) is a well-known Bayesian algorithm that has been extensively analyzed in both multi-armed bandit problems (Kaufmann et al., 2012; Agrawal & Goyal, 2012) and RL (Osband et al., 2013). Osband & Van Roy (2017) provides a regret rate $\tilde{O}(H\sqrt{SAK})$ for vanilla Thompson sampling in RL, which is called the PSRL algorithm. Recently, other works adapted Thompson sampling to hierarchical Bayesian problems (Hong et al., 2020; 2022a;b; Kveton et al., 2021). Mixture Thompson sampling (Hong et al., 2022b), which is similar to PSRL but samples the unknown MDP from a mixture prior, is arguably the closest to our setting. In this paper, we take inspiration from their algorithm to design C-PSRL and derive its analysis, even though, instead of their tabular setting, we tackle a fundamentally different problem on factored MDPs resulting from a casual graph prior, which induces unique challenges.

**Factored MDPs.** Previous works considered RL in FMDPs (Boutilier et al., 2000) with either known (Osband & Van Roy, 2014; Xu & Tewari, 2020; Talebi et al., 2021; Tian et al., 2020; Chen et al., 2020) or unknown (Strehl et al., 2007; Vigorito & Barto, 2009; Diuk et al., 2009; Chakraborty & Stone, 2011; Hallak et al., 2015; Guo & Brunskill, 2017; Rosenberg & Mansour, 2021) factorization. The PSRL algorithm has been adapted to both finite-horizon (Osband & Van Roy, 2014) and infinite-horizon (Xu & Tewari, 2020) FMDPs. The former assumes knowledge of the factorization, close to our setting with an oracle prior, and provides Bayesian regret of order $\tilde{O}(Hd_Y^{3/2}N^{Z/2}\sqrt{K})$. Previous works also studied RL in FMDPs in a frequentist sense, either with known (Chen et al., 2020) or unknown (Rosenberg & Mansour, 2021) factorization. Rosenberg & Mansour (2021) employ an optimistic method that is orthogonal to ours, whereas they leave as an open problem capturing the effect of prior knowledge, for which we provide answers in a Bayesian setting.

**Causal RL.** Various works addressed RL with a causal perspective (see Kaddour et al., 2022, Chapter 7). Causal principles are typically exploited to obtain compact representations of states and transitions (Tomar et al., 2021; Gasse et al., 2021), or to pursue generalization across tasks and environments (Zhang et al., 2020; Huang et al., 2022; Feng et al., 2022; Mutti et al., 2023). Closer to

our setting, Lu et al. (2022) aim to exploit prior causal knowledge to learn in both MDPs and FMDPs. Our work differs from theirs in two key aspects: We show how to exploit a partial causal graph prior instead of assuming knowledge of the full causal graph, and we consider a Bayesian formulation of the problem while they tackle a frequentist setting through optimism principles. Zhang (2020b) show an interesting application of causal RL for designing treatments in a DTR problem.

**Causal bandits.** Another research line connecting causal models and sequential decision-making is the one on *causal bandits* (Lattimore et al., 2016; Sen et al., 2017; Lee & Bareinboim, 2018; 2019; Lu et al., 2020; 2021; Nair et al., 2021; Xiong & Chen, 2022; Feng & Chen, 2023), in which the actions of the bandit problem correspond to interventions on variables of a causal graph. There, the causal model specifies a particular structure on the actions, modelling their dependence with the rewarding variable, instead of the transition dynamics as in our work. Moreover, they typically assume the causal model to be known, with the exception of (Lu et al., 2021), and they study the simple regret in a frequentist sense rather than the Bayesian regret given a partial causal prior.

## 8 CONCLUSION

In this paper, we presented how to exploit prior knowledge expressed through a partial causal graph to improve the statistical efficiency of reinforcement learning. Before reporting some concluding remarks, it is worth commenting on where such a causal graph prior might be originated from.

**Exploiting experts' knowledge.** One natural application of our methodology is to exploit domain-specific knowledge coming from experts. In several domains, e.g., medical or scientific applications, expert practitioners have some knowledge over the causal relationships between the domain's variables. However, they might not have a full picture of the causal structure, especially when they face complex systems such as the human body or biological processes. Our methodology allows those practitioners to easily encode their partial knowledge into a graph prior, instead of having to deal with technically involved Bayesian statistics to specify parametric prior distributions, and then let C-PSRL figure out a competent decision policy with the given information.

**Exploiting causal discovery.** Identifying the causal graph over domain's variables, which is usually referred as causal discovery, is a main focus of *causality* (Pearl, 2009, Chapter 3). The literature provides plenty of methods to perform causal discovery from data (Peters et al., 2017, Chapter 4), including learning causal variables and their relationships in MDP settings (Zhang et al., 2020; Mutti et al., 2023). However, learning the full causal graph, even when it is represented with a bigraph as in MDP settings (Mutti et al., 2023), can be statistically costly or even prohibitive (Gillispie & Perlman, 2001; Wadhwa & Dong, 2021). Moreover, not all the causal edges are guaranteed to transfer across environments (Mutti et al., 2023), which would force to perform causal discovery anew for any slight variation of the domain (e.g., changing the patient in a DTR setting). Our methodology allows to focus on learning the *universal* causal relationships (Mutti et al., 2023), which transfer across environments, e.g., different patients, and then specify the prior through a partial causal graph.

The latter paragraphs describe two scenarios in which our work enhance the applicability of PSRL, bridging the gap between how the prior might be specified in practical applications and what previous methods currently require, i.e., a parametric prior distribution. To summarize our contributions, we first provided a Bayesian formulation of reinforcement learning with prior knowledge expressed through a partial causal graph. Then, we presented an algorithm, C-PSRL, tailored for the latter problem, and we analyzed its regret to obtain a rate that is sublinear in the number of episodes and shows a direct dependence with the degree of causal knowledge. Finally, we derived an ancillary result to show that C-PSRL embeds a notion of causal discovery, and we provided an empirical validation of the algorithm against relevant baselines. C-PSRL resulted nearly competitive with F-PSRL, which enjoys a richer prior, while clearly outperforming PSRL with an uninformative prior.

Future works may derive a tighter analysis of the Bayesian regret of C-PSRL, as well as a stronger causal discovery result that allows to recover a minimal causal graph instead of a super-graph. Another important aspect is to address computational issues inherent to planning in FMDPs to scale the implementation of C-PSRL to complex domains. Finally, interesting future directions include extending our framework to model-free PSRL (Dann et al., 2021; Tiapkin et al., 2023), in which the prior may specify causal knowledge of the reward or the value function directly, and to study how prior misspecification (Simchowitz et al., 2021) affects the regret rate.

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

APPENDIX

## A    LIST OF SYMBOLS

**Basic mathematical objects**

| | | |
|---|---|---|
| $\mathcal{A}$ | $\triangleq$ | Set or space |
| $A$ | $\triangleq$ | Constant or random variable |
| $a$ | $\triangleq$ | Element of a set |
| $\Delta(\mathcal{A})$ | $\triangleq$ | Probability simplex over $\mathcal{A}$ |
| $f : \mathcal{A} \to \mathcal{B}$ | $\triangleq$ | Function from $\mathcal{A}$ to $\mathcal{B}$ |
| $[A]$ | $\triangleq$ | Set of integers $[A] = \{1, \ldots, A\}$ |
| $x[\mathcal{I}]$ | $\triangleq$ | Scope operator $x[\mathcal{I}] := \bigotimes_{i \in \mathcal{I}} x_i$ for any set $\mathcal{I} \subseteq [d]$, $x \in \mathbb{R}^d$ |

**Causal graph**

| | | |
|---|---|---|
| $\mathcal{G}$ | $\triangleq$ | Directed acyclic bigraph $\mathcal{G} = (\mathcal{X}, \mathcal{Y}, z)$ |
| $\mathcal{X}$ | $\triangleq$ | Set of $d_X$ random variables $\{X_j\}_{j=1}^{d_X}$ taking values $x_j \in [N]$ |
| $\mathcal{Y}$ | $\triangleq$ | Set of $d_Y$ random variables $\{Y_j\}_{j=1}^{d_Y}$ taking values $y_j \in [N]$ |
| $z$ | $\triangleq$ | Directed edges $z \subseteq \mathcal{X} \times \mathcal{Y}$ |
| $z_j$ | $\triangleq$ | Parents of $Y_j$ such that $z_j = \{i \mid (X_i, Y_j) \in z\}$ |
| $Z$ | $\triangleq$ | Degree of sparseness such that $|z_j| < Z, \forall j \in [d_Y]$ |
| $N$ | $\triangleq$ | Size of the support of random variables |

**MDP**

| | | |
|---|---|---|
| $\mathcal{M}$ | $\triangleq$ | Markov decision process $\mathcal{M} = (\mathcal{S}, \mathcal{A}, p, r, \mu, H)$ |
| $\mathcal{S}$ | $\triangleq$ | State space |
| $\mathcal{A}$ | $\triangleq$ | Action space |
| $p$ | $\triangleq$ | Transition model $p : \mathcal{S} \times \mathcal{A} \to \Delta(\mathcal{S})$ |
| $r$ | $\triangleq$ | Reward function $r : \mathcal{S} \times \mathcal{A} \to \Delta([0, 1])$ |
| $\mu$ | $\triangleq$ | Initial state distribution $\mu \in \Delta(\mathcal{S})$ |
| $H$ | $\triangleq$ | Episode horizon $H < \infty$ |
| $S$ | $\triangleq$ | Size of the state space $S = |\mathcal{S}|$ |
| $A$ | $\triangleq$ | Size of the action space $A = |\mathcal{A}|$ |
| $s$ | $\triangleq$ | State $s \in \mathcal{S}$ |
| $a$ | $\triangleq$ | Action $a \in \mathcal{A}$ |
| $R(s, a)$ | $\triangleq$ | Mean reward $\mathbb{E}[r(s, a)]$ |

**Factored MDP**

| | | |
|---|---|---|
| $\mathcal{F}$ | $\triangleq$ | Factored Markov Decision Process $\mathcal{F} = (\{\mathcal{X}_j\}_{j=1}^{d_X}, \{\mathcal{Y}_j, z_j, p_j, r_j, \}_{j=1}^{d_Y}, \mu, H, Z, N)$ |
| $d_X$ | $\triangleq$ | Number of state-action variables |
| $d_Y$ | $\triangleq$ | Number of state variables |
| $\mathcal{X}$ | $\triangleq$ | Factored state-action space $\mathcal{X} = \mathcal{X}_1 \times \ldots \times \mathcal{X}_{d_X}$ |
| $\mathcal{Y}$ | $\triangleq$ | Factored state space $\mathcal{Y} = \mathcal{Y}_1 \times \ldots \times \mathcal{Y}_{d_Y}$ |
| $z$ | $\triangleq$ | Directed edges $z \subseteq \mathcal{X} \times \mathcal{Y}$, i.e., a *factorization* |
| $z_j$ | $\triangleq$ | Parents of $Y_j$ such that $z_j = \{i \mid (X_i, Y_j) \in z\}$ |
| $p$ | $\triangleq$ | Factored transition model $p(y|x) = \prod_{j=1}^{d_Y} p_j(y[j] \mid x[z_j])$ |
| $r$ | $\triangleq$ | Factored reward function $r(x) = \sum_{j=1}^{d_Y} r_j(x[z_j])$ |
| $\mu$ | $\triangleq$ | Initial state distribution $\mu \in \Delta(\mathcal{Y})$ |
| $H$ | $\triangleq$ | Episode horizon $H < \infty$ |
| $Z$ | $\triangleq$ | Degree of sparseness such that $|z_j| < Z, \forall j \in [d_Y]$ |
| $N$ | $\triangleq$ | Size of the support of state and action variables |

**Learning problem**

| | | |
|---|---|---|
| $K$ | $\triangleq$ | Number of episodes |
| $k$ | $\triangleq$ | Episode index |
| $h$ | $\triangleq$ | Step index |

| | | |
|---|---|---|
| $\mathcal{Z}$ | $\triangleq$ | Space of the consistent factorizations $\mathcal{Z} \subseteq \mathcal{X} \times \mathcal{Y}$ |
| $\mathcal{Z}_j$ | $\triangleq$ | Space of the consistent parents of $Y_j$ such that $\mathcal{Z} = \mathcal{Z}_1 \times \ldots \mathcal{Z}_{d_Y}$ |
| $P_0$ | $\triangleq$ | Hyper-prior on the factorizations consistent with $\mathcal{G}_0$ (supported in $\mathcal{Z}$) |
| $P_k$ | $\triangleq$ | Posterior of the hyper-prior $P_0$ at episode $k \in [K]$ |
| $P_0(\cdot|z)$ | $\triangleq$ | Prior on the FMDPs with factorization $z$ |
| $P_k(\cdot|z)$ | $\triangleq$ | Posterior of the prior $P_0(\cdot|z)$ at episode $k \in [K]$ |
| $P_{\mathcal{G}_0}$ | $\triangleq$ | Prior on the FMDPs consistent with $\mathcal{G}_0$ such that |

$$P_{\mathcal{G}_0}(\mathcal{F}) = P_0(p_\mathcal{F}|z_\mathcal{F})P_0(z_\mathcal{F})$$

| | | |
|---|---|---|
| $\mathcal{BR}(K)$ | $\triangleq$ | $K$-episodes Bayesian regret |

**Regret analysis**

| | | |
|---|---|---|
| $\Omega$ | $\triangleq$ | Set of all the possible assignments of $X = \{X_i\}_{i \in [d_X]}$, $\Omega = \bigotimes_{i \in [d_X]}[N]$ |
| $n$ | $\triangleq$ | Index on the support of random variables, $n \in [N]$ |
| $\mathcal{H}_k$ | $\triangleq$ | History of observations $((x_{h,l}, r_{h,l}))_{h \in [H], l \in [k-1]}$ until episode $k$ |
| $Z_k$ | $\triangleq$ | Random variable of the global factorization at episode $k$ |
| $Z_j^k$ | $\triangleq$ | Random variable of the local factorization at episode $k$ for factor $j$ |
| $Z_*$ | $\triangleq$ | Random variable of the true factorization |
| $Z_{j*}$ | $\triangleq$ | Random variable of the true factorization for $j$-th factor |
| $\mathbb{E}_k[\cdot]$ | $\triangleq$ | Conditional expectation given history $\mathcal{H}_k$, $\mathbb{E}_k[\cdot] := \mathbb{E}[\cdot \mid \mathcal{H}_k]$ |
| $\mathbb{P}_k[\cdot]$ | $\triangleq$ | Conditional probability given history $\mathcal{H}_k$, $\mathbb{P}_k[\cdot] := \mathbb{P}[\cdot \mid \mathcal{H}_k]$ |

## B PARAMETRIC PRIORS AND POSTERIOR UPDATES

In the following, we detail how the hyper-priors and priors of C-PSRL (Algorithm 1) can be specified through parametric distributions, and how the corresponding parameters are updated with the evidence provided by the collected data.

The hyper-prior $P_0 = \{P_{0,j}\}_{j=1}^{d_Y}$ is defined through $d_Y$ distributions over the set of *local factorizations* $\mathcal{Z}_1, \ldots, \mathcal{Z}_{d_Y}$ resepctively, where each $\mathcal{Z}_j$ contains the parents assignments for the variable $Y_j$ consistent with the graph prior $\mathcal{G}_0$. Let assume any arbitrary ordering of the local factorizations $z_i \in \mathcal{Z}_j$, such that each $z_i$ is indexed by $i \in [|\mathcal{Z}_j|]$. Then, we can specify the hyper-prior $j$ as a categorical distribution

$$P_{0,j}(z_i; \boldsymbol{\omega}) = \mathrm{Cat}(i; \boldsymbol{\omega}) = \frac{\omega_i}{\sum_t \omega_t},$$

where the sum is over $t \in [|\mathcal{Z}_j|]$, and the vector of parameters $\boldsymbol{\omega}$ is initialized as $\boldsymbol{\omega} = (1, \ldots, 1)$.

Then, for each local factorization $z_i \in \mathcal{Z}_j$ of the variable $Y_j$, we specify the prior $P_{0,j}(\cdot | z_i)$ over the model parameters of the corresponding transition factor $p_j$. The transition factor $p_j$ is an $(N^{|\mathcal{Z}_j|}, N)$ stochastic matrix. The prior is specified through a Dirichlet distribution for each row of $p_j$, i.e.,

$$p_{0,j}(\cdot \mid z_i; \boldsymbol{\alpha}) = \mathrm{Dir}(\theta_1, \ldots, \theta_N; \alpha_1, \ldots, \alpha_N) = \frac{1}{B(\boldsymbol{\alpha})} \prod_{n=1}^{N} \theta_n^{\alpha_n - 1},$$

where $\boldsymbol{\alpha}$ is a vector of parameters initialized as $\boldsymbol{\alpha} = (1, \ldots, 1)$ and $B(\alpha) = \prod_{n=1}^{N} \Gamma(\alpha_n)/\Gamma(\sum_n \alpha_n)$ is a normalizing factor.

Having specified the hyper-prior and prior, we now show how to update them with the new evidence. Let be $\theta_1, \ldots, \theta_N \sim P_{k,j}(\cdot | x[z_j]; \boldsymbol{\alpha})$, and assume to collect the transition $(x[z_j], y[j] = i)$ from the true FMDP $\mathcal{F}_*$. Then, the posterior is

$$P_{k+1,j}(\theta_1, \ldots, \theta_N) \propto P(y[j] = i \mid \theta_1, \ldots, \theta_N) P_{k,j}(\theta_1, \ldots, \theta_N; \boldsymbol{\alpha}) \propto \theta_i \prod_{n=1}^{N} \theta^{\alpha_n - 1},$$

which is still a Dirichlet distribution with parameters $\mathrm{Dir}(\theta_1, \ldots, \theta_N; \alpha_1, \ldots, \alpha_i + 1, \ldots, \alpha_N)$. Then, we can propagate the posterior up the hierarchy to update the hyper-prior as

$$P_{k+1,j}(z) \propto P(y[j] = i \mid z) P_{t,j}(z; \boldsymbol{\omega})$$

$$\propto P_{k,j}(z; \boldsymbol{\omega}) \int P(y[j] = i \mid \theta_1, \ldots, \theta_N) P_{t,j}(\theta_1, \ldots, \theta_N \mid z) \mathrm{d}(\theta_1, \ldots, \theta_N) \quad (1)$$

$$\propto P_{k,j}(z; \boldsymbol{\omega}) \int \theta_i \frac{1}{B(\boldsymbol{\alpha})} \prod_{n=1}^{N} \theta_n^{\alpha_n - 1} \mathrm{d}(\theta_1, \ldots, \theta_N) \quad (2)$$

$$\propto P_{k,j}(z; \boldsymbol{\omega}) \frac{1}{B(\boldsymbol{\alpha})} B(\alpha_1, \ldots, \alpha_i + 1, \ldots, \alpha_N) \quad (3)$$

$$\propto P_{k,j}(z; \boldsymbol{\omega}) \frac{\alpha_i + 1}{\sum_t \alpha_t + 1} \quad (4)$$

where (2) is obtained by plugging the parametric prior in (1), we derive (3) by computing the integral over the simplex of $(\theta_1, \ldots, \theta_N)$, and (4) follows from $\Gamma(\alpha_i + 1) \prod_{t \neq i} \Gamma(\alpha_t) = (\alpha_i + 1) \prod_t \Gamma(\alpha_t)$ and $\Gamma(\alpha_i + 1 + \sum_{t \neq i} \alpha_t) = \Gamma(\sum_t \alpha_t) \sum_t (\alpha_t + 1)$. The resulting posterior is still a categorical distribution with the parameters $\omega_i \leftarrow \omega_i \frac{\alpha_i + 1}{\sum_t \alpha_t + 1}$ .

## C    NOTE ON COMPUTATIONAL COMPLEXITY

As we mentioned in Section 3 (paragraph on "Planning in FMDPs"), the C-PSRL algorithm is not fully tractable as it requires to solve exact planning in a FMDP (line 6). As we pointed out in the paper, the computational issue may be overcome through clever approximation schemes (Guestrin et al., 2003) or exploiting the structure of the specific FMDP instance. E.g., under linear realizability of the transition model or the value function, which means the transition and value function can be expressed as linear combinations of a vector of features, tractable exact planning methods have been developed (Yang & Wang, 2019; Jin et al., 2020b; Deng et al., 2022). Furthermore, this computational issue is not specific to the C-PSRL algorithm but affects PSRL in FMDPs in general.

Nonetheless, C-PSRL actually induces an additional computational cost over the standard F-PSRL algorithm. This is the burn-in cost of computing the set of consistent factorizations $\mathcal{Z}$ in line 2 of Algorithm 1. Notably, our method allows to build the set of consistent parents for each variable $Y_j$ independently (see Section 3 "Recipe"), which means the process can be parallelized on $d_Y$ workers. For each worker, we need to build a set of $\sum_{i=0}^{Z-\eta} \binom{d_X - \eta}{i}$ elements, which we can do recursively by calling the base function at most $O(2^{d_X - \eta})$ times. The latter result gracefully characterize how the degree of prior knowledge $\eta$ impacts the statistical (see Theorem 4.1) and computational complexity in a similar way.

While we underline again that this is not the computational bottleneck of C-PSRL, understanding how to avoid the computational burn-in (e.g., not pre-computing the whole set of factorizations but building it incrementally) is a nice direction for future works.

# D   WEAK CAUSAL DISCOVERY

In the following, we show that we can extract, under a relatively mild causal minimality assumption, a $Z$-sparse super-DAG of the true causal graph $\mathcal{G}_{\mathcal{F}_*}$ as a byproduct of a run of Algorithm 1. We call this result *weak causal discovery*, to make a clear distinction between discovering a sparse super-DAG of a causal graph and true causal discovery, in which the minimal graph is discovered.

As required for any causal discovery algorithm, we need to state an assumption that connects the causal graph $\mathcal{G}_{\mathcal{F}_*}$ with the distribution $p_*$ (i.e., the transition model) from which our observations are sampled in an i.i.d. manner (Spirtes et al., 2000). Typically, in causal discovery, it is assumed that $p_*$ fulfills the faithfulness assumption with regard to $\mathcal{G}_{\mathcal{F}_*}$, i.e., every independence in $p_*$ implies a $d$-separation (see Definition 4 below) in $\mathcal{G}_{\mathcal{F}_*}$. Faithfulness, however, is a rather strong assumption which can be violated by path cancellations or xor-type dependencies, and weaker assumptions have been proposed (Spirtes et al., 2000; Pearl, 2009; Marx et al., 2021). In this work, we build upon a strictly weaker assumption than faithfulness: causal minimality (Spirtes et al., 2000).[7]

**Definition 3** (Causal Minimality). *A distribution $P$ satisfies causal minimality with respect to a DAG $\mathcal{G}$ if $P$ fulfills the Markov factorization property with respect to $\mathcal{G}$, but not with respect to any proper subgraph of $\mathcal{G}$.*

**Intuition.**   More intuitively, a distribution is minimal with respect to $\mathcal{G}$ if and only if there is no node that is conditionally independent of any of its parents, given the remaining parents (Peters et al., 2017). There are two important points in this statement: i) none of the parents of a node is redundant, and ii) the dependence to a parent may only be detected given the remaining parents. Aspect ii) is a strictly weaker statement than required by faithfulness, which can be illustrated with a simple example. Consider the causal structure $X \to Y \leftarrow Z$, where all random variables are binary. If we generate $X$ and $Z$ via an unbiased coin and assign $Y$ as $Y := X$ xor $Z$, $Y$ will be marginally independent of $X$, as well as marginally independent of $Z$. However, $Y$ is not independent of $X$ (resp. $Z$) when we condition on its second parent $Z$ (resp. $X$). Such an example violates faithfulness, i.e., there is a causal edge that is not matched by a dependence, but it does not violate causal minimality. For a more detailed discussion on such triples, we refer to (Marx et al., 2021).

In our context, Algorithm 1 has a positive probability of sampling all parents jointly (or a superset of them), and does not rely on checking pairs individually. Therefore, we can build upon the weaker assumption, causal minimality. Beyond identifiability in the limit, we are interested in the finite sample behaviour of our approach. Therefore, we propose a slightly stronger assumption for the value function, which is inspired by causal minimality.

**Definition 2** ($\epsilon$-Value Minimality). *An FMDP $\mathcal{F}$ fulfills $\epsilon$-value minimality, if for any FMDP $\mathcal{F}'$ encoding a proper subgraph of $\mathcal{G}_{\mathcal{F}}$, i.e., $\mathcal{G}_{\mathcal{F}'} \subset \mathcal{G}_{\mathcal{F}}$, it holds that $V_{\mathcal{F}}^* > V_{\mathcal{F}'}^* + \epsilon$, where $V_{\mathcal{F}}^*$, $V_{\mathcal{F}'}^*$ are the value functions of the optimal policies in $\mathcal{F}$, $\mathcal{F}'$ respectively.*

Intuitively, $\epsilon$-value minimality ensures that if we were to miss a true parent, the resulting optimal value function would be at most $\epsilon$-optimal compared to the optimal value function evaluated on a graph that contains all true parents. Based on this rather lightweight assumption, we can extract from Algorithm 1 a graph $\mathcal{G}_{\mathcal{F}_K}$ that is guaranteed to be either the true DAG $\mathcal{G}_{\mathcal{F}_*}$, or a $Z$-sparse super-DAG of $\mathcal{G}_{\mathcal{F}_*}$ with high probability.

**Corollary 5.1** (Weak Causal Discovery). *Let $\mathcal{F}_*$ be an FMDP in which the transition model $p_*$ fulfills the causal minimality assumption with respect to $\mathcal{G}_{\mathcal{F}_*}$, and let $\mathcal{F}_*$ fulfill $\epsilon$-value minimality. Then, $\mathcal{G}_{\mathcal{F}_*} \subseteq \mathcal{G}_{\mathcal{F}_K}$ holds with high probability, where $\mathcal{G}_{\mathcal{F}_K}$ is a $Z$-sparse graph randomly selected within the sequence $\{\mathcal{G}_{\mathcal{F}_k}\}_{k=0}^{K-1}$ produced by C-PSRL over $K = \tilde{O}(H^5 d_Y^2 2^{d_X - \eta} / \epsilon^2)$ episodes.*

*Proof.* From Theorem 4.1, we have that the $K$-episodes Bayesian regret of Algorithm 1 is

$$\mathbb{E}\left[ \sum_{k=0}^{K-1} V_*(\pi_*) - V_*(\pi_t) \right] \leq C_1 \sqrt{H^5 d_Y^2 2^{d_X - \eta}} \cdot \sqrt{K},$$

---

[7]The definition refers SGS-minimality proposed by Spirtes et al. (2000). There exists an alternative definition called P-minimality, proposed by Pearl (2009). In our setting, both assumptions are equivalent, since they only differ on graphs that violate triangle faithfulness (Zhang, 2020a; Zhang & Spirtes, 2008). Since no nodes within $\mathcal{X}$ or within $\mathcal{Y}$ are allowed to be adjacent, such triangle structures cannot occur within our assumptions.

with high probability for some constant $C_1$ that does not depend on $K$. Through a standard regret-to-pac argument (Jin et al., 2018), it follows

$$\mathbb{E}\left[V_*(\pi_*) - V_*(\pi_K)\right] \leq C_2\sqrt{H^5 d_Y^2 2^{d_X - \eta}} \cdot \frac{1}{\sqrt{K}} \tag{5}$$

with high probability for some constant $C_2$ that does not depend on $K$, and for a policy $\pi_K$ that is randomly selected within the sequence of policies $\{\pi_k\}_{k=0}^{K-1}$ produced by Algorithm 1. By noting that $\pi_K$ can be $\epsilon$-optimal in the true FMDP $\mathcal{F}_*$ only if $\mathcal{G}_{\mathcal{F}_*} \subseteq \mathcal{G}_{\mathcal{F}_K}$ through the $\epsilon$-value minimality assumption (Definition 2), we let $\mathbb{E}\left[V_*(\pi_*) - V_*(\pi_K)\right] = \epsilon$ in (5), which gives $K \geq C_2 H^5 d_Y^2 2^{d_X - \eta}/\epsilon^2$ and concludes the proof. $\qquad\square$

$d$**-Separation.** For the reader's convenience, here we report a brief definition of $d$-separation. More details can be found in (Peters et al., 2017).

**Definition 4** ($d$-Separation). *A path $\langle X, \ldots, Y \rangle$ between two vertices $X, Y$ in a DAG is d-connecting given a set $\boldsymbol{Z}$, if*

1. *every collider[8] on the path is an ancestor of $\boldsymbol{Z}$, and*

2. *every non-collider on the path is not in $\boldsymbol{Z}$.*

*If there is no path d-connecting $X$ and $Y$ given $\boldsymbol{Z}$, then $X$ and $Y$ are d-separated given $\boldsymbol{Z}$. Sets $\boldsymbol{X}$ and $\boldsymbol{Y}$ are d-separated given $\boldsymbol{Z}$, if for every pair $X, Y$, with $X \in \boldsymbol{X}$ and $Y \in \boldsymbol{Y}$, $X$ and $Y$ are d-separated given $\boldsymbol{Z}$.*

---

[8]A collider $C$ on a path $\langle \ldots, Q, C, W, \ldots \rangle$ is a node with two arrowhead pointing towards it, i.e. $\rightarrow C \leftarrow$.

# E   REGRET ANALYSIS

In this section, we provide the full derivation of the following result.

**Theorem 4.1.** *Let $\mathcal{G}_0$ be a causal graph prior with degree of sparseness $Z$ and degree of prior knowledge $\eta$. The $K$-episodes Bayesian regret incurred by* C-PSRL *is*

$$\mathcal{BR}(K) = \tilde{O}\left(\left(H^{5/2}N^{1+Z/2}d_Y + \sqrt{H2^{d_X-\eta}}\right)\sqrt{K}\right).$$

On a high level, the proof is made up of two parts. The first part (presented in Section E.1) consists of decomposing the Bayesian regret into two components and then upper bounding the two expressions separately. This leads to the intermediate regret bound for a general latent hypothesis space, i.e., where the hypothesis space is not necessarily a product space, reported in Section E.1. The second part of the proof refines the analysis by considering a product latent hypothesis space (Section E.2) and the degree of prior knowledge (Section E.3), ultimately reaching the theorem statement.

We define the set $\Omega = \bigotimes_{i\in[d_X]}[N]$ of all the possible assignments of $X = \{X_i\}_{i\in[d_X]}$. For the sake of concision, we will denote $p_k\left(y[j] \mid x[z_j]\right)$ as $p_k\left(x[z_j]\right)$ where it will not lead to ambiguity. Moreover, we denote $\mathbb{E}_k[\cdot] := \mathbb{E}[\cdot \mid \mathcal{H}_k]$ and $\mathbb{P}_k[\cdot] := \mathbb{P}[\cdot \mid \mathcal{H}_k]$ the conditional expectation and probability given the history of observations $\mathcal{H}_k = ((x_{h,l}, r_{h,l}))_{h\in[H],l\in[k-1]}$ collected until episode $k$. Auxiliary results and lemmas mentioned alongside the analysis are reported in the Sections E.4 and E.5.

## E.1   ANALYSIS FOR A GENERAL LATENT HYPOTHESIS SPACE

We first report a decomposition of the Bayesian regret and then proceeds to bound each component separately, which are then combined in a single regret rate.

**Bayesian regret decomposition.**   For episode $k$, we define $\overline{V}_k(\pi, z) = \mathbb{E}_{\mathcal{F}\sim P_k(\cdot|z)}[V_{\mathcal{F}}(\pi)]$ as the expected value of policy $\pi$ according to the posterior conditioned on the latent factorization $z \sim P_k$ and history $\mathcal{H}_k$. As shown in (Russo & Van Roy, 2014, Proposition 1) for the bandit setting and in (Hong et al., 2022b, Section 5.1, Equation 6) for the reinforcement learning setting, we can decompose the Bayesian regret as

$$\mathcal{BR}(n) = \mathbb{E}\left[\sum_{k=1}^K \mathbb{E}_k\left[V_*(\pi_*) - \overline{V}_k(\pi_*, Z_*)\right]\right] + \mathbb{E}\left[\sum_{k=1}^K \mathbb{E}_k\left[\overline{V}_k(\pi_k, Z_k) - V_*(\pi_k)\right]\right] \quad (6)$$

by adding and subtracting $\overline{V}_k(\pi_*, Z_*)$ and noticing that $\pi_*, Z_*$ are identically distributed to $\pi_k, Z_k$ given $\mathcal{H}_k$. Notice that $Z_k$ and $Z_*$ indicate random variables, while we will indicate with the low-ercase counterpart specific values of these random variables. The first term represents the regret incurred due to the concentration of the posteriors of the reward and transition models given the true factorization, while the second term captures the cost to identify the true latent factorization. We will bound each term of (6) separately.

**Upper bounding the first term of** (6).   For episode $k$, we define the event

$$E_k = \left\{\forall\, j \in [d_Y], \forall\, x[z_j] \in \Omega : \left|R_{\mathcal{F}_k}(x[z_j]) - \bar{r}_k(x[z_j^k])\right| \le c_k(x[z_j^k])\right.$$

$$\left. \text{and } \|p_{\mathcal{F}_k}(x[z_j]) - \bar{p}_k(x[z_j^k])\|_1 \le \phi(x[z_j^k])\right\}$$

where the quantities are defined as follows. $\mathcal{F}_k$ denotes the FMDP sampled at episode $k$ having mean reward $R_{\mathcal{F}_k}(x[z_j])$ and transition model $p_{\mathcal{F}_k}(x[z_j])$ for all $x[z_j] \in \Omega$. The expression $\bar{r}_k(x[z_j]) = \mathbb{E}_{\mathcal{F}\sim P_k(\cdot|z)}[R_{\mathcal{F}}(x[z_j])]$ denotes the posterior mean of $R_{\mathcal{F}_k}(x[z_j])$, while $\bar{p}_k(x[z_j]) = (\bar{p}_k(y[j] = n \mid x[z_j]))_{n\in[N]}$ with $\bar{p}_k(y[j] = n \mid x[z_j]) = \mathbb{E}_{\mathcal{F}\sim P_k(\cdot|z)}[p_{\mathcal{F}}(y[j] = n \mid x[z_j])]$ denotes the posterior mean transition probability vector of size $N$ for the $j$-th factor given a factorization $z$. With $c_k(x[z_j^k])$ and $\phi(x[z_j^k])$ we denote high-probability confidence widths for the $j$-th factor of the mean reward and transition model respectively. A detailed derivation of such confidence widths can be found in Section E.4. Informally, the event $E_k$ expresses how close the mean rewards and

transition models sampled at the episode $k$ are to their posterior means. We refer with $\bar{E}_k$ to the complementary event of $E_k$.

Now, by reminding that $\pi_*, Z_*$ are identically distributed to $\pi_k, Z_k$ given $\mathcal{H}_k$, we can rewrite each element of the sum within the first term of (6) as

$$\underset{k}{\mathbb{E}} \left[ V_{\mathcal{F}_k}(\pi_k) - \overline{V}_k(\pi_k, Z_k) \right]$$

$$\overset{(1)}{=} \underset{k}{\mathbb{E}} \left[ \underset{\mathcal{F} \sim P_k(\cdot | Z_k)}{\mathbb{E}} \left[ V_{\mathcal{F}_k}(\pi_k) - V_{\mathcal{F}}(\pi_k) \right] \right]$$

$$\overset{(2)}{\leq} \underset{k}{\mathbb{E}} \left[ \sum_{h=1}^{H} \left( R_{\mathcal{F}_k}(S_{k,h}, A_{k,h}) - \bar{r}_k(S_{k,h}, A_{k,h}, Z_k) \right) + H \| p_{\mathcal{F}_k}(S_{k,h}, A_{k,h}) - \bar{p}_k(S_{k,h}, A_{k,h}, Z_k) \|_1 \right]$$

$$\overset{(3)}{\leq} \underset{k}{\mathbb{E}} \left[ \sum_{h=1}^{H} \sum_{j=1}^{d_Y} \left( R_{\mathcal{F}_k}(X_{k,h}[Z_j]) - \bar{r}_k(X_{k,h}[Z_j^k]) \right) + H \sum_{j=1}^{d_Y} \| p_{\mathcal{F}_k}(X_{k,h}[Z_j]) - \bar{p}_k(X_{k,h}[Z_j^k]) \|_1 \right]$$

$$\leq \underset{k}{\mathbb{E}} \left[ H \sum_{h=1}^{H} \sum_{j=1}^{d_Y} \left( R_{\mathcal{F}_k}(X_{k,h}[Z_j]) - \bar{r}_k(X_{k,h}[Z_j^k]) \right) + \| p_{\mathcal{F}_k}(X_{k,h}[Z_j]) - \bar{p}_k(X_{k,h}[Z_j^k]) \|_1 \right]$$

$$\overset{(4)}{\leq} \underset{k}{\mathbb{E}} \left[ H \sum_{h=1}^{H} \sum_{j=1}^{d_Y} \left( R_{\mathcal{F}_k}(X_{k,h}[Z_j]) - \bar{r}_k(X_{k,h}[Z_j^k]) + \| p_{\mathcal{F}_k}(X_{k,h}[Z_j]) - \bar{p}_k(X_{k,h}[Z_j^k]) \|_1 \right) \mathbb{1}\{\bar{E}_k\} \right]$$

$$+ \underset{k}{\mathbb{E}} \left[ H \sum_{h=1}^{H} \sum_{j=1}^{d_Y} \left( c_k(X_{k,h}[Z_j^k]) + \phi_k(X_{k,h}[Z_j^k]) \right) \mathbb{1}\{E_k\} \right] \tag{7}$$

where we have used the definition of $\overline{V}_k(\pi_k, Z_k)$ in step (1), Lemma E.3 in step (2), Lemma E.4 in step (3), and the definition of $E_k$ in step (4).

By defining $\beta_k(X_{k,h}[Z_j^k]) := c_k(X_{k,h}[Z_j^k]) + \phi_k(X_{k,h}[Z_j^k])$ as the sum of both confidence widths, we can bound the second term of (7) by using Lemma E.5, while we bound the first term of the same equation by showing that $\bar{E}_k$ conditioned on $\mathcal{H}_k$ is unlikely. We rewrite the first term of (6) as

$$H \sum_{h=1}^{H} \sum_{j=1}^{d_Y} \left( \underset{k}{\mathbb{E}} \left[ \left( R_{\mathcal{F}_k}(X_{k,h}[Z_j]) - \bar{r}_k(X_{k,h}[Z_j^k]) \right) \mathbb{1}\{\bar{E}_k\} \right] \right.$$

$$\left. + \underset{k}{\mathbb{E}} \left[ \| p_{\mathcal{F}_k}(X_{k,h}[Z_j]) - \bar{p}_k(X_{k,h}[Z_j^k]) \|_1 \mathbb{1}\{\bar{E}_k\} \right] \right) \tag{8}$$

where we have distributed the indicator function. For the first term within the sums of (8), we have

$$\underset{k}{\mathbb{E}} \left[ \left( R_{\mathcal{F}_k}(X_{k,h}[Z_j]) - \bar{r}_k(X_{k,h}[Z_j^k]) \right) \mathbb{1}\{\bar{E}_k\} \right] \tag{9}$$

$$\leq \sum_{x[Z_j] \in \Omega} \int_{r=c_k(X_{k,h}[Z_j^k])}^{\infty} r \, \mathbb{P}_k \left( \left( R_{\mathcal{F}_k}(x[Z_j]) - \bar{r}_k(x[Z_j^k]) \right) = r \right) dr \tag{10}$$

$$\leq \sum_{x[Z_j] \in \Omega} \mathbb{P}_k \left( R_{\mathcal{F}_k}(x[Z_j]) - \bar{r}_k(x[Z_j^k]) \geq c_k(x[Z_j^k]) \right) \tag{11}$$

$$\overset{(1)}{\leq} \sum_{x[Z_j] \in \Omega} \exp \left( -\frac{c_k(x[Z_j^k])^2}{2/4(\|\alpha_k^R(x[Z_j^k])\|_1 + 1)} \right) \tag{12}$$

$$\overset{(2)}{=} \sum_{x[Z_j] \in \Omega} \exp \left( -\frac{\frac{\log(2K d_Y N^Z)}{2(\|\alpha_k^R(x[Z_j^k])\|_1 + 1)}}{2/4(\|\alpha_k^R(x[Z_j^k])\|_1 + 1)} \right) \tag{13}$$

$$= \sum_{x[Z_j] \in \Omega} \exp \left( -\log(2K d_Y N^Z) \right) = \frac{1}{2K d_Y} \tag{14}$$

In step (1) we have used Lemma E.2 and E.1, and in step (2) we have plugged-in the definition of $c_k(x[Z_j^k])$ from (plugged-in $\sigma^2$ of $R_\Delta$), where $\alpha_k^R(x[Z_j^k])$ represents the parameters of the posterior over the mean reward for the $j$-th factor at episode $k$ given factorization $Z_j^k$. For the second term within the sums of (8), we have

$$
\begin{aligned}
&\mathbb{E}_k\left[\|p_{\mathcal{F}_k}(X_{k,h}[Z_j]) - \bar{p}_k(X_{k,h}[Z_j^k])\|_1 \mathbb{1}\{\bar{E}_k\}\right] \\
&\overset{(1)}{\leq} N \mathbb{E}_k\left[\max_{n\in[N]} |p_{\mathcal{F}_k}(X_{k,h}[Z_j], n) - \bar{p}_k(X_{k,h}[Z_j^k], n)| \mathbb{1}\{\bar{E}_k\}\right] \\
&\overset{(2)}{\leq} \sum_{x[Z_j]\in\Omega} \sum_{n\in[N]} \int_{p=\phi_k(X_{k,h}[Z_j^k])/\sqrt{N}}^{\infty} p\, \mathbb{P}_k(|p_{\mathcal{F}_k}(X_{k,h}[Z_j], n) - \bar{p}_k(X_{k,h}[Z_j^k], n)| = p)\, dp \\
&\leq 2\,\mathbb{P}_k\left(|p_{\mathcal{F}_k}(X_{k,h}[Z_j], n) - \bar{p}_k(X_{k,h}[Z_j^k], n)| \geq \frac{\phi_k(x[Z_j^k])}{\sqrt{N}}\right) \\
&\overset{(3)}{\leq} \sum_{x[Z_j]\in\Omega} \sum_{n\in[N]} 2\exp\left(-\frac{\phi_k(x[Z_j^k])^2}{2N/4(\|\alpha_k^P(x[Z_j^k])\|_1+1)}\right) \\
&= \frac{N}{Kd_Y}
\end{aligned}
\tag{15}
$$

The steps are analogous to the ones for upper bounding the first term of (8). Specifically, in step (1) we use a trivial upper bound on the $l^1$-norm and in step (2) we divide the confidence width by the square root of the vector length $\sqrt{N}$ according to lemma (Lattimore & Szepesvári, 2020, Theorem 5.4.c). The parameters $\alpha_k^P(x[Z_j^k])$ introduced in step (3) represent the parameters of the posterior over the transition model for the $j$-th factor at episode $k$ given factorization $Z_j^k$.

By plugging (14) and (15) into (8) and then (8) into (7), we can bound the first term of (6) as

$$
\mathbb{E}\left[\sum_{k=1}^{K} \mathbb{E}_k\left[V_*(\pi_*) - \overline{V}_k(\pi_*, Z_*)\right]\right] \leq NH^2 + H\sum_{k=1}^{K}\sum_{h=1}^{H}\sum_{j=1}^{d_Y} \mathbb{E}_k\left[\beta_k(X_{k,h}[Z_j^k])\right]
\tag{16}
$$

where we recall that $\beta_k(X_{k,h}[Z_j^k]) := c_k(X_{k,h}[Z_j^k]) + \phi_k(X_{k,h}[Z_j^k])$.

**Upper bounding the second term of** (6). Since there is no fundamental distinction between latent states in the tabular and factored MDP settings, our analysis in this section is aligned with (Hong et al., 2022b, Appendix B.3, step 2) and aims at effectively translating it into the factored MDPs notation.

In order to bound the second term of (6), we first need to define confidence sets over latent factorizations. For each episode $k$, we define a set of factorizations $C_k$ so that $Z_* \in C_k$ with high probability. Since the latent factorization is unobserved, we can only exploit a proxy statistic for how well the model parameter posterior of each latent factorization predicts the rewards. We start defining a counting function $N_k(z) = \sum_{l=1}^{k-1} \mathbb{1}\{Z_l = z\}$ as the number of times the factorization $z$ has been sampled until episode $k$. Next, we define the following statistic associated with a factorization $z$ and episode $k$,

$$
G_k(z) = \sum_{l=1}^{k-1} \mathbb{1}\{Z_l = z\}\left(\overline{V}_l(\pi_l, z) - H\sqrt{2}\sum_{h=1}^{H}\sum_{j=1}^{d_Y} \beta_l(X_{l,h}[z_j]) - \sum_{h=0}^{H-1}\sum_{j=1}^{d_Y} R_{l,h}[j]\right)
$$

The latter represents the total under-estimation of observed returns, as it expresses the difference between the lower confidence bound on the returns and the observed ones, assuming that $z$ is the true latent factorization. Now we can define $C_k = \{z \in \mathcal{Z} : G_k(z) \leq \sqrt{HN_k(z)\log K}\}$ as the set of latent factorizations with at most $\sqrt{HN_k(z)\log K}$ excess. In the following, we show that $Z_* \in C_k$ holds with high probability for any episode.

Fix $Z_* = z$. Let $\mathcal{T}_{k,z} = \{l < k : Z_l = z\}$ the set of episodes where $z$ has been sampled until episode $k$. We will first upper bound $G_k(z)$ by a martingale with respect to the history, then bound

the martingale using Azuma-Hoeffding's inequality. We define the event

$$\mathcal{E}_{k,h,j} = \Bigg\{ \left| R_{\mathcal{F}_k}(X_{k,h}[Z_j]) - \bar{r}_k(X_{k,h}[Z_j^k]) \right| \leq \sqrt{2} c_k(X_{k,h}[Z_j^k])$$

$$\text{and } \| p_{\mathcal{F}_k}(X_{k,h}[Z_j]) - \bar{p}_k(X_{k,h}[Z_j^k]) \|_1 \leq \sqrt{2} \phi_k(X_{k,h}[Z_j^k]) \Bigg\}$$

in which the sampled reward and transition probabilities for factor $j$ in step $h$ of episode $k$ are close to their posterior means. Let $\mathcal{E} = \cup_{k=1}^{K} \cup_{h=1}^{H} \cup_{j=1}^{d_Y} \mathcal{E}_{k,h,j}$ be the event that this holds for every factor, step, and episode. By union bound we have that

$$\mathbb{P}_k(\bar{\mathcal{E}}_{k,h,j}) \leq \mathbb{P}_k\left( \left| R_{\mathcal{F}_k}(X_{k,h}[Z_j]) - \bar{r}_k(X_{k,h}[Z_j^k]) \right| \geq \sqrt{2} c_k(X_{k,h}[Z_j^k]) \right)$$

$$+ \mathbb{P}_k(\| p_{\mathcal{F}_k}(X_{k,h}[Z_j]) - \bar{p}_k(X_{k,h}[Z_j^k]) \|_1 \geq \sqrt{2} \phi_k(X_{k,h}[Z_j^k]))$$

$$\overset{(1)}{\leq} \exp\left( \frac{2 c_k(X_{k,h}[Z_j^k])^2}{\sigma^2} \right) + \exp\left( \frac{2 \phi_k(X_{k,h}[Z_j^k])^2}{N \sigma^2} \right)$$

$$\leq (K d_Y N^{d_X})^{-2}$$

where we have used Lemmas E.2 and E.1 in step (1) and $\bar{\mathcal{E}}_{k,h,j}$ is the complementary event of $\mathcal{E}_{k,h,j}$. Hence, for $\bar{\mathcal{E}} = \cup_{k=1}^{K} \cup_{h=1}^{H} \cup_{j=1}^{d_Y} \bar{\mathcal{E}}_{k,h,j}$, we have

$$\mathbb{P}(\bar{\mathcal{E}}) = \sum_{k=1}^{K} \sum_{h=1}^{H} \sum_{z \in \mathcal{Z}} \sum_{j=1}^{d_Y} \sum_{x[Z_j] \in \Omega} \mathbb{P}_k(\bar{\mathcal{E}}_{k,h,j}) \leq \sum_{k=1}^{K} \sum_{h=1}^{H} \sum_{z \in \mathcal{Z}} \sum_{j=1}^{d_Y} \sum_{x[Z_j] \in \Omega} (K d_Y N^{d_X})^{-2} \leq H |\mathcal{Z}| K^{-1}$$

For episode $l \in \mathcal{T}_{k,z}$, let $\Delta_l = V_*(\pi_l) - \sum_{h=1}^{H} \sum_{j=1}^{d_Y} R_{l,h}[j]$. Since $\mathbb{E}_l[\Delta_l] = 0$, $(\Delta_l)_{l \in \mathcal{T}_{k,z}}$ is a martingale difference sequence with respect to the histories $(\mathcal{H}_l)_{l \in \mathcal{T}_{k,z}}$. Following exactly the same steps as in (Hong et al., 2022b, Proof of Lemma 7), we derive an upper bound on the probability of $Z_*$ not being in the factorizations set $C_k$, namely:

$$\mathbb{P}(Z_* \notin C_k) \leq 2 |\mathcal{Z}| H K^{-1} \tag{17}$$

We can now decompose the second term of (6) according to whether the sampled latent factorization is in $C_k$ or not. Formally, we have

$$\mathbb{E}\left[ \sum_{k=1}^{K} \mathbb{E}_k \left[ \overline{V}_k(\pi_k, Z_k) - V_*(\pi_k) \right] \right] \leq \mathbb{E}\left[ \sum_{k=1}^{K} \left( \overline{V}_k(\pi_k, Z_k) - V_*(\pi_k) \right) \mathbb{1}\{Z_k \in C_k\} \right]$$

$$+ H \sum_{k=1}^{K} \mathbb{P}(Z_* \notin C_k) \tag{18}$$

From the previous steps, using (17), we have that the second term of (18) is upper bounded by $2|\mathcal{Z}|H^2$, while in the following we derive an upper bound for the first term of (18) as in (Hong et al., 2022b, Appendix B.3, step 4). We have

$$\mathbb{E}\left[ \sum_{k=1}^{K} \left( \overline{V}_k(\pi_k, Z_k) - V_*(\pi_k) \right) \mathbb{1}\{Z_k \in C_k\} \right]$$

$$= H \sqrt{2} \, \mathbb{E}\left[ \sum_{k=1}^{K} \sum_{h=1}^{H} \sum_{j=1}^{d_Y} \beta_k(X_{k,h}[Z_j^k]) \right]$$

$$+ \mathbb{E}\left[ \sum_{k=1}^{K} \left( \overline{V}_k(\pi_k, Z_k) - H\sqrt{2} \sum_{h=1}^{H} \sum_{j=1}^{d_Y} \beta_k(X_{k,h}[Z_j^k]) - \sum_{h=1}^{H} \sum_{j=1}^{d_Y} R_{k,h}[j] \right) \mathbb{1}\{Z_k \in C_k\} \right]$$

$$\overset{(1)}{\leq} H\sqrt{2} \, \mathbb{E}\left[ \sum_{k=1}^{K} \sum_{h=1}^{H} \sum_{j=1}^{d_Y} \beta_k(X_{k,h}[Z_j^k]) \right] + \mathbb{E}\left[ \sum_{z \in \mathcal{Z}} G_{K+1}(z) + |\mathcal{Z}| H \right]$$

$$\overset{(2)}{\le} H\sqrt{2}\,\mathbb{E}\left[\sum_{k=1}^{K}\sum_{h=1}^{H}\sum_{j=1}^{d_Y}\beta_k(X_{k,h}[Z_j^k])\right] + \sqrt{|\mathcal{Z}|KH\log K} + |\mathcal{Z}|H$$

where in step (1) we use the definition of $G_{K+1}(z)$ and in step (2) we upper bound the same quantity.

**Bayesian regret for a general latent hypothesis space.** Combining the upper bounds of the two terms of (6), we get

$$\mathcal{BR}(K) \le NH^2 + H\sum_{k=1}^{K}\sum_{h=1}^{H}\sum_{j=1}^{d_Y}\mathbb{E}_k\left[\beta_k(X_{k,h}[Z_j^k])\right] + H\sqrt{2}\,\mathbb{E}\left[\sum_{k=1}^{K}\sum_{h=1}^{H}\sum_{j=1}^{d_Y}\beta_k(X_{k,h}[Z_j^k])\right]$$

$$+ \sqrt{|\mathcal{Z}|KH\log K} + |\mathcal{Z}|H + 2|\mathcal{Z}|H^2$$

$$\overset{(1)}{\le} NH^2 + 3H^2 d_Y N^{d_x} + 3H^2 d_Y N\sqrt{N^{d_x} KH\log(4Kd_Y N^{d_x})\log\left(1 + \frac{KH}{2N^{d_x}\Lambda_{0,z}}\right)}$$

$$+ \sqrt{|\mathcal{Z}|KH\log K} + 3|\mathcal{Z}|H^2$$

where in step (1) we have used Lemma E.5, and we denote

$$\Lambda_{0,z} = \min\left\{\min_{j,x[z_j]}\|\alpha_0^R(x[z_j])\|_1, \min_{j,x[z_j]}\|\alpha_0^P(x[z_j])\|_1\right\}$$

Due to the $Z$-sparseness assumption, we can rewrite the Bayesian regret as

$$\mathcal{BR}(K) \le NH^2 + 3H^2 d_Y N^Z + 3H^2 d_Y N\sqrt{N^Z KH\log(4Kd_Y N^Z)log\left(1 + \frac{KH}{2N^Z\Lambda_{0,z}}\right)}$$

$$+ \sqrt{|\mathcal{Z}|KH\log K} + 3|\mathcal{Z}|H^2$$

$$= \tilde{O}\left(H^2 d_Y N^Z + H^{\frac{5}{2}} d_Y N^{1+\frac{Z}{2}}\sqrt{K} + \sqrt{|\mathcal{Z}|KH} + |\mathcal{Z}|H^2\right)$$

Notably, this rate is sublinear in the number of episodes $K$ and latent factorization $|\mathcal{Z}|$, exponential in the degree of sparseness $Z$.

## E.2 REFINEMENT 1: PRODUCT LATENT HYPOTHESIS SPACES

As we briefly explained in Section 3, C-PSRL samples the factorization $z$ from the product space $\mathcal{Z} = \mathcal{Z}_1 \times \ldots \times \mathcal{Z}_{d_Y}$ by combining independent samples $z_j \in \mathcal{Z}_j$ for each variable $Y_j$. This allows us to refine the dependence in $|\mathcal{Z}|$ to $\overline{C} := \max_{j\in[d_Y]}|\mathcal{Z}_j| \le |\mathcal{Z}|$. We can replicate the same steps of the previous section in order to derive the Bayesian regret for the setting with a product latent hypothesis space. For the sake of clarity, we report here the main steps highlighting the difference with the previous section.

For an episode $k \in [K]$, we define $C_k^j = \left\{z_j \in \mathcal{Z}_j : G_k^j(\bar{z}) \le \sqrt{HN_k^j(z_j)\log K}\right\}$ where

$$G_k^j(z_j) = \sum_{l=1}^{k-1}\mathbb{1}\{Z_j^l = z_j\}H\sum_{h=1}^{H}\Big(\big(\bar{r}_l(X_{l,h}[z_j]) - R_{l,h}[j]\big)$$

$$+ \|\bar{p}_k(X_{l,h}[z_j]) - p_*(X_{l,h}[z_{j*}])\|_1 - \sqrt{2}\beta_k(X_{l,h}[z_j])\Big)$$

and $N_k^j(z_j) = \sum_{l=1}^{k-1}\mathbb{1}\{Z_j^l = z_j\}$. While $G_k^j(z_j)$ captures the under-estimation of the observed returns at the level of a single factor, $N_k^j(z_j)$ counts the number of times that the local factorization $z_j$ has been sampled for node $j$ until episode $k$. Next, we define $\mathcal{T}_{k,z_j}^j = \{l < k : Z_j^l = z_j\}$ as the set of episodes where $z_j$ has been sampled for node $j$.

First, we can derive an upper bound of $\mathbb{P}(\bar{\mathcal{E}})$ depending on $\overline{C}$ by noticing that the inner-most sum depends only on the local factorization hence we can swap the two preceding sums over $\mathcal{Z}$ and $d_Y$

as shown in step (1) of the following:

$$\mathbb{P}(\bar{\mathcal{E}}) = \sum_{k=1}^{K}\sum_{h=1}^{H}\sum_{z\in\mathcal{Z}}\sum_{j=1}^{d_Y}\sum_{x[z_j]\in\Omega}\mathbb{P}_k(\bar{\mathcal{E}}_{k,h,j})$$

$$\leq \sum_{k=1}^{K}\sum_{h=1}^{H}\sum_{z\in\mathcal{Z}}\sum_{j=1}^{d_Y}\sum_{x[z_j]\in\Omega}(Kd_Y N^{d_X})^{-2}$$

$$\overset{(1)}{=}\leq \sum_{k=1}^{K}\sum_{h=1}^{H}\sum_{j=1}^{d_Y}\sum_{z_j\in\mathcal{Z}_j}\sum_{x[z_j]\in\Omega}(Kd_Y N^{d_X})^{-2}$$

$$\leq H\overline{C}K^{-1}$$

Now, we wish to upper bound $\mathbb{P}(Z_{j*}\notin C_k^j)$. For episode $l\in\mathcal{T}_{k,z_j}^j$ let $\Delta_l^j = \sum_{h=1}^{H}R_*(X_{l,h}[Z_{j*}]) - \sum_{h=1}^{H}R_{l,h}[j]$. Since $\mathbb{E}_l[\Delta_l^j] = 0$ we have that $(\Delta_l^j)_{l\in\mathcal{T}_{t,z_j}^j}$ is a martingale difference sequence with respect to the histories $(\mathcal{H}_l)_{l\in\mathcal{T}_{t,z_j}^j}$.

By following the same steps as in Hong et al. (2022b), we get

$$G_k^j(z_j)\mathbb{1}\{\mathcal{E}\} = \sum_{l\in\mathcal{T}_{t,z_j}^j}H\sum_{h=1}^{H}\Big(\big(\bar{r}_l(X_{l,h}[z_j]) - R_{l,h}[j]\big)$$

$$+ \|\bar{p}_k(X_{l,h}[z_j]) - p_*(X_{l,h}[z_{j*}])\|_1 - \sqrt{2}\beta_k(X_{l,h}[z_j])\Big) \leq \sum_{l\in\mathcal{T}_{k,z_j}^j}\Delta_l^j$$

and by fixing $|\mathcal{T}_{t,z_j}^j| = N_t^j(z_j) = u < t$, and using Azuma-Hoeffding's inequality, we derive

$$\mathbb{P}_k\left(G_k^j(\bar{z})\mathbb{1}\{\mathcal{E}\} \geq \sqrt{Hu\log K}\right) \leq \mathbb{P}\left(\sum_{l\in\mathcal{T}_{k,z_j}^j}\Delta_l^j \geq \sqrt{Hu\log K}\right) \leq K^{-2}.$$

Therefore, by using union bounds, we can write

$$\mathbb{P}\left(Z_*\notin C_k\right) \leq \sum_{j=1}^{d_Y}\mathbb{P}(Z_{j*}\notin C_k^j)$$

$$\leq \sum_{j=1}^{d_Y}\sum_{z_j\in\mathcal{Z}_j}\sum_{u=1}^{k-1}\mathbb{P}\left(G_k^j(z_j) \geq \sqrt{Hu\log K}\right)$$

$$\leq \mathbb{P}(\bar{\mathcal{E}}) + \sum_{j=1}^{d_Y}\sum_{z_j\in\mathcal{Z}_j}\sum_{u=1}^{k-1}\mathbb{P}\left(G_k^j(z_j)\mathbb{1}\{\mathcal{E}\} \geq \sqrt{Hu\log K}\right)$$

$$\leq d_Y H\overline{C}K^{-1} \tag{19}$$

We can now decompose the second term of (6) according to whether the sampled latent factorization is in $C_k$ or not, as in the previous section. Formally, we have

$$\mathbb{E}\left[\sum_{k=1}^{K}\mathbb{E}_k\left[\overline{V}_k(\pi_k, Z_k) - V_*(\pi_k)\right]\right] \leq \mathbb{E}\left[\sum_{k=1}^{K}\left(\overline{V}_k(\pi_k, Z_k) - V_*(\pi_k)\right)\mathbb{1}\{Z_k\in C_k\}\right]$$

$$+ H\sum_{k=1}^{K}\mathbb{P}(Z_*\notin C_k)$$

From (19), we know that the second term is upper bounded by $d_Y\overline{C}H^2$. Meanwhile, we can bound the first term as follows

$$\mathbb{E}\left[\sum_{k=1}^{K}\left(\overline{V}_k(\pi_k, Z_k) - V_*(\pi_k)\right)\mathbb{1}\{Z_k\in C_k\}\right]$$

$$\leq \mathbb{E}\left[ \sum_{k=1}^{K} H \sum_{h=1}^{H} \sum_{j=1}^{d_Y} \left( \bar{r}_k(X_{k,h}[Z_j^k]) - R_*(X_{k,h}[Z_{j*}]) \right. \right.$$

$$\left. \left. + \|\bar{p}_k(X_{l,h}[Z_j^k]) - p_*(X_{l,h}[Z_{j*}])\|_1 \right) \mathbb{1}\{Z_j^k \in C_k^j\} \right]$$

$$= H\sqrt{2}\,\mathbb{E}\left[ \sum_{k=1}^{K} \sum_{h=1}^{H} \sum_{j=1}^{d_Y} \beta_k(X_{k,h}[Z_j^k]) \right] + \mathbb{E}\left[ \sum_{k=1}^{K} H \sum_{h=1}^{H} \sum_{j=1}^{d_Y} \left( \bar{r}_k(X_{k,h}[Z_j^k]) - R_*(X_{k,h}[Z_{j*}]) \right. \right.$$

$$\left. \left. + \|\bar{p}_k(X_{l,h}[Z_j^k]) - p_*(X_{l,h}[Z_{j*}])\|_1 - \sqrt{2}\beta_k(X_{k,h}[Z_j^k]) \right) \mathbb{1}\{Z_j^k \in C_k^j\} \right]$$

$$\overset{(1)}{\leq} H\sqrt{2}\,\mathbb{E}\left[ \sum_{k=1}^{K} \sum_{h=1}^{H} \sum_{j=1}^{d_Y} \beta_k(X_{k,h}[Z_j^k]) \right] + \mathbb{E}\left[ \sum_{j=1}^{d_Y} \sum_{z_j \in \mathcal{Z}_j} G_{K+1}^j(z_j) + d_Y \overline{C} H \right]$$

$$\overset{(2)}{\leq} H\sqrt{2}\,\mathbb{E}\left[ \sum_{k=1}^{K} \sum_{h=1}^{H} \sum_{j=1}^{d_Y} \beta_k(X_{k,h}[Z_j^k]) \right] + \sum_{j=1}^{d_Y} \sum_{z_j \in \mathcal{Z}_j} \frac{1}{d_Y}\sqrt{HN_{K+1}^j(z_j)\log K} + d_Y \overline{C} H$$

$$\leq H\sqrt{2}\,\mathbb{E}\left[ \sum_{k=1}^{K} \sum_{h=1}^{H} \sum_{j=1}^{d_Y} \beta_k(X_{k,h}[Z_j^k]) \right] + \sqrt{\overline{C}KH\log K} + d_Y \overline{C} H$$

where in step (1) we use the definition of $G_{K+1}^j(z)$ and in step (2) we upper bound the same quantity.

**Bayesian regret for a product latent hypothesis space.** Exploiting the $Z$-sparseness assumption, we can write

$$\mathcal{BR}(K) = \tilde{O}\left( H^2 d_Y N^Z + H^{\frac{5}{2}} d_Y N^{1+\frac{Z}{2}}\sqrt{K} + \sqrt{\overline{C}KH} + d_Y \overline{C} H^2 \right) \tag{20}$$

Notably, this rate is sublinear in the number of episodes $K$ and the number of latent local factorizations $\overline{C}$, exponential in the degree of sparseness $Z$.

### E.3 REFINEMENT 2: DEGREE OF PRIOR KNOWLEDGE

Finally, we aim to capture the dependency in the degree of prior knowledge $\eta$ in the Bayesian regret. To do that, we have to express $\overline{C} = \max_{j \in [d_Y]} |\mathcal{Z}_j|$ in terms of $\eta$. We can write

$$\overline{C} = \sum_{i=0}^{Z} \binom{d_X}{i} = \sum_{i=0}^{Z} C_i^{d_X}$$

where we count the empty factorization when $i = 0$. Given a graph hyper-prior that fixes $\eta < Z$ edges for each node $j \in [d_Y]$, we can count the number of admissible local factorizations as

$$\overline{C} = \sum_{i=0}^{Z-\eta} \binom{d_X - \eta}{i}$$

where we count the factorization with only the edges fixed a priori when $i = 0$. We can build an upper bound on $\overline{C}$ as follows.

$$\overline{C} = \sum_{i=0}^{Z-\eta} \binom{d_X - \eta}{i}$$

$$\leq 2^{d_X - \eta - 1} \exp\left( \frac{(d_X + \eta - 2Z - 2)^2}{4(1 + Z - d_X)} \right)$$

$$\leq 2^{d_X - \eta} \exp\left( \frac{(d_X + \eta - 2Z)^2}{4(1 + Z - d_X)} \right) =: \phi(d_X, Z, \eta)$$

Since it is hard to interpret the rate of the latter upper bound, we derive a looser version that is easier to interpret. We have

$$\overline{C} = \sum_{i=0}^{Z-\eta} \binom{d_X - \eta}{i}$$
$$= 2^{d_X - \eta} - \sum_{i=0}^{d_X - Z - 1} \binom{d_X - \eta}{i}$$
$$\leq 2^{d_X - \eta} - 2^{d_X - Z} + 1$$
$$\leq 2^{d_X - \eta}$$

From the latter we can notice that each unit of the degree of prior knowledge $\eta$ make the hypothesis space shrink with an exponential rate, and thus the corresponding regret terms as well. In particular, by plugging-in the upper bound $\overline{C} \leq 2^{d_X - \eta} = \frac{2^{d_X}}{2^\eta}$ in the Bayesian regret in (20), we obtain the final upper bound, which is

$$\mathcal{BR}(K) = \tilde{O}\left(H^2 d_Y N^Z + H^{\frac{5}{2}} d_Y N^{1+\frac{Z}{2}}\sqrt{K} + \sqrt{2^{d_X - \eta} KH} + d_Y 2^{d_X - \eta} H^2\right). \quad (21)$$

### E.4 HIGH PROBABILITY CONFIDENCE WIDTHS

Here we define high-probability confidence widths on the reward function and transition model along the lines of Hong et al. (2022b), but with the difference that the confidence widths are defined for all factors and their possible assignments rather than for state-action pairs as in the tabular setting. We denote as $c_k(x[z_j])$ and $\phi_k(x[z_j])$ the confidence widths for the $j$-th factor of the reward function and the transition model respectively. In the following, we indicate with $R_\mathcal{F}$ and $p_\mathcal{F}$ the mean reward and transition model of the FMDP $\mathcal{F}$ respectively.

**Reward function.** First, we write the posterior mean reward for the $j$-th factor, given a factorization $z$ as $\bar{r}_k(x[z_j]) = \mathbb{E}_{\mathcal{F} \sim P_k(\cdot|z)}[R_\mathcal{F}(x[z_j])]$. We wish to have a high probability bound of the type

$$\mathbb{P}_k\left(|R_\mathcal{F}(x[z_j]) - \bar{r}_k(x[z_j])| \geq c_k(x[z_j])\right) \leq \frac{1}{K}$$

for all $j \in [d_Y]$ and possible assignments $x[z_j] \in \Omega$. By the union bound, we have

$$\mathbb{P}_k\left(\left|R_\mathcal{F}(x[z_j]) - \bar{r}_k(x[z_j])\right| \geq c_k(x[z_j])\right)$$
$$= \mathbb{P}_k\left(\bigcup_{j=1}^{d_Y} \bigcup_{x[z_j] \in \Omega} \left\{|R_\mathcal{F}(x[z_j]) - \bar{r}_k(x[z_j])| \geq c_k(x[z_j])\right\}\right)$$
$$\leq \sum_{j=1}^{d_Y} \sum_{x[z_j] \in \Omega} \mathbb{P}_k\left(|R_\mathcal{F}(x[z_j]) - \bar{r}_k(x[z_j])| \geq c_k(x[z_j])\right).$$

Applying a union bound again to the latter expression, we can derive the following one-sided bound:

$$\mathbb{P}_k\left(R_\mathcal{F}(x[z_j]) - \bar{r}_k(x[z_j]) \geq c_k(x[z_j])\right) \leq \frac{1}{2K d_Y N^{d_X}}.$$

According to Lemma E.1, $R_\Delta := R_\mathcal{F}(x[z_j]) - \bar{r}_k(x[z_j])$ is a $\sigma^2$-subgaussian random variable with $\sigma^2 = 1/\left(4\left(\|\alpha_k^R(z_j)\|_1 + 1\right)\right)$. Therefore, through the Cramèr-Chernoff method exploited in Lemma E.2, we have that the high probability bound above holds if

$$\exp\left(-\frac{c_k(x[z_j])^2}{2\sigma^2}\right) \leq \frac{1}{2K d_Y N^{d_X}}$$

which holds if and only if

$$
c_k(x[z_j]) \geq \sqrt{2\sigma^2 \log(2K d_Y N^{d_X})}
$$

$$
= \sqrt{\frac{\log(2K d_Y N^{d_X})}{2\left(\|\alpha_k^R(x[z_j])\|_1 + 1\right)}} \qquad \text{(plugged-in } \sigma^2 \text{ of } R_\Delta\text{)}
$$

Hence, we pick $c_k(x[z_j]) := \sqrt{\frac{\log(2K d_Y N^Z)}{2\left(\|\alpha_k^R(x[z_j])\|_1 + 1\right)}}$, where $Z$ is a lower bound on the value of $d_X$, which holds due to the $Z$-sparseness assumption.

**Transition model.** The derivation is analogous to the one for the reward function, hence here we report only the differences. First, we write the posterior mean transition probability for the $j$-th factor, given a factorization $z$ as $\bar{p}_k(y[j] = n \mid x[z_j]) = \mathbb{E}_{\mathcal{F} \sim P_k(\cdot|z)}[p_{\mathcal{F}}(y[j] = n \mid x[z_j])]$, which is the probability, according to the posterior at time $k$, over the element $n \in [N]$ of the domain of the $j$-th component of $Y$. Since we want to bound the deviations over all components of a factor, we define the vector form of the previous expression as $\bar{p}_k(x[z_j]) = (\bar{p}_k(y[j] = n \mid x[z_j]))_{n \in [N]}$. By following the same steps as for the confidence width of the reward function, we get

$$
\phi_k(x[z_j]) := \sqrt{\frac{N \log(2K d_Y N^Z)}{2\left(\|\alpha_k^P(x[z_j])\|_1 + 1\right)}} \tag{22}
$$

where the $\sqrt{N}$ term is due to (Lattimore & Szepesvári, 2020, Theorem 5.4.c), since the $l^1$ norm sums over $N\sigma^2$-subgaussian random variables and therefore the induced random variable is $(N\sigma^2)$-subgaussian.

## E.5 Auxiliary lemmas

**Lemma E.1** (Theorem 1 and 3 of Marchal & Arbel (2017)). *Let $X \sim \text{Beta}(\alpha, \beta)$ for $\alpha, \beta > 0$. Then $X - \mathbb{E}[X]$ is $\sigma^2$-subgaussian with $\sigma^2 = 1/(4(\alpha + \beta + 1))$. Similarly, let $X \sim \text{Dir}(\alpha)$ for $\alpha \in \mathbb{R}_+^d$. Then $X - \mathbb{E}[X]$ is $\sigma^2$-subgaussian with $\sigma^2 = 1/\left(4\left(\|\alpha\|_1 + 1\right)\right)$.*

**Lemma E.2** (Theorem 5.3 of Lattimore & Szepesvári (2020)). *If $X$ is $\sigma^2$-subgaussian, then for any $\varepsilon \geq 0$,*

$$
\mathbb{P}(X \geq \varepsilon) \leq \exp\left(-\frac{\varepsilon^2}{2\sigma^2}\right)
$$

**Lemma E.3** (Value Difference Lemma, Lemma 6 Hong et al. (2022b)). *For any MDPs $\mathcal{M}', \mathcal{M}$, and policy $\pi$,*

$$
V_{\mathcal{M}'}(\pi) - V_{\mathcal{M}}(\pi) \leq \mathop{\mathbb{E}}_{\mathcal{M}}\left[\sum_{h=1}^{H} R_{\mathcal{M}'}(S_h, A_h) - R_{\mathcal{M}}(S_h, A_h) + H \|p_{\mathcal{M}'}(S_h, A_h) - p_{\mathcal{M}}(S_h, A_h)\|_1\right]
$$

*Proof.* This upper bound can be obtained by trivially upper bounding with 1 the reward at each step, and therefore with $h$ the value function within the statement in (Jin et al., 2020a, Lemma C.1). $\square$

**Lemma E.4** (Deviations of Factored Reward and Transitions Osband & Van Roy (2014)). *Given two reward functions $R$ and $\bar{R}$ with scopes $\{z_j\}_{j=1}^{d_Y}$ we can upper bound the deviations by*

$$
|R(x) - \bar{R}(x)| \leq \sum_{j=1}^{d_Y} |R_j(x[z_j]) - \bar{R}_j(x[z_j])|
$$

*and, given two transition models $p$ and $\bar{p}$ with scopes $\{z_j\}_{j=1}^{d_Y}$ we can upper bound the deviations by*

$$
|p(x) - \bar{p}(x)| \leq \sum_{j=1}^{d_Y} |p_j(x[z_j]) - \bar{p}_j(x[z_j])|
$$

**Lemma E.5.** *For episode $k$ and latent factorization $z \in \mathcal{Z}$, let $\beta_k(x[z_j]) = c_k(x[z_j]) + \phi_k(x[z_j])$ for any $j \in [d_Y]$ and $x[z_j] \in \Omega$. Let $\Lambda_{0,z} = \min\left\{\min_{j,x[z_j]} \|\alpha_0^R(x[z_j])\|_1, \min_{j,x[z_j]} \|\alpha_0^P(x[z_j])\|_1\right\}$ indicates the minimum level of concentration between the reward function and transition model priors for any factor and latent factorization $z$. Then, we have*

$$\sum_{k=1}^{K}\sum_{h=1}^{H}\sum_{j=1}^{d_Y}\beta_k(X_{k,h}[z_j]) \le Hd_Y N^{d_X} + d_Y NH\sqrt{N^{d_X}KH\log(4Kd_Y N^{d_X})\log\left(1 + \frac{KH}{2N^{d_X}\Lambda_{0,z}}\right)}$$

*Proof.* We define $N_k(x[z_j]) = \sum_{l=1}^{k-1}\sum_{h=1}^{H}\mathbb{1}\{X_{l,h}[z_j] = x[z_j]\}$ as the number of times the assignment $x[z_j]$ was sampled up to episode $k$ for factor $j$. We can decompose the sum as

$$\sum_{k=1}^{K}\sum_{h=1}^{H}\sum_{j=1}^{d_Y}\beta_k(X_{k,h}[z_j])$$

$$\le \sum_{k=1}^{K}\sum_{h=1}^{H}\sum_{j=1}^{d_Y}\mathbb{1}\{N_k(X_{k,h}[z_j]) \le h\} + \sum_{k=1}^{K}\sum_{h=1}^{H}\sum_{j=1}^{d_Y}\mathbb{1}\{N_k(X_{k,h}[Z_j]) > H\}\beta_k(X_{k,h}[z_j])$$

where we upper bound by 1 the regret in a step due to one factor. Therefore, the first term is upper bounded by $Hd_Y N^{d_X}$ since there are at most $N^{d_X}$ assignments for each $j \in [d_Y]$ and the same one can appear in the sum at most $H$ times, thus removing the dependency on $K$. Due to the assumption of $Z$-sparseness, we will later use the bound $Hd_Y N^Z$. As for the second term, we define $N_{k,h}(x[z_j]) = N_k(x[z_j]) + \sum_{p=1}^{h-1}\mathbb{1}\{X_{k,p}[z_j] = x[z_j]\}$ as the number of times $x[z_j]$ was sampled up to step $h$ of episode $k$, for factor $j$. We split $\beta_k$ into $c_k$ and $\phi_k$. For $c_k$ we have:

$$\sum_{k=1}^{K}\sum_{h=1}^{H}\sum_{j=1}^{d_Y}\mathbb{1}\{N_k(X_{k,h}[z_j]) > H\}c_k(X_{k,h}[z_j])$$

$$= \sum_{k=1}^{K}\sum_{h=1}^{H}\sum_{j=1}^{d_Y}\mathbb{1}\{N_k(X_{k,h}[z_j]) > H\}\sqrt{\frac{\log(2Kd_Y N^Z)}{2\left(\|\alpha_k^R(X_{k,h}[z_j])\|_1 + 1\right)}}$$

$$= \sum_{k=1}^{K}\sum_{h=1}^{H}\sum_{j=1}^{d_Y}\sum_{x[z_j]\in\Omega}\mathbb{1}\{N_k(x[z_j]) > H\}\sqrt{\frac{\log(2Kd_Y N^Z)}{2\|2\alpha_k^R(X_{k,h}[z_j])\|_1 + 2N_k(x[z_j]) + 2}}$$

$$\le \sum_{k=1}^{K}\sum_{h=1}^{H}\sum_{j=1}^{d_Y}\sum_{x[z_j]\in\Omega}\sqrt{\frac{\log(2Kd_Y N^Z)}{2\|2\alpha_k^R(X_{k,h}[z_j])\|_1 + N_{k,h}(x[z_j])}}$$

$$\le \sqrt{\log(2Kd_Y N^Z)}\sum_{j=1}^{d_Y}\sum_{x[z_j]\in\Omega}\sqrt{N_{K+1}(x[z_j])\sum_{u=1}^{N_{K+1}(x[z_j])}\frac{1}{2\|\alpha_k^R(X_{k,h}[z_j])\|_1 + u}}$$

$$\le \sqrt{N^{d_X}KH\log(2Kd_Y N^Z)}\sum_{j=1}^{d_Y}\sqrt{\sum_{u=1}^{KH/N^{d_X}}\frac{1}{2\Lambda_{0,z} + u}}$$

$$\le d_Y\sqrt{N^{d_X}KH\log(2Kd_Y N^Z)\log\left(1 + \frac{KH}{2N^{d_X}\Lambda_{0,z}}\right)}$$

where in the first step we have plugged-in $c_k$ as picked in Section 4, in the second step have exploited the posterior update rule, in step three we have used that if $N_k(x[z_j]) > H$ we have that $N_{k,h}(x[z_j]) \le N_k(x[z_j]) + H \le 2N_k(x[z_j])$, and in the remaining passages we have used known bounds as in (Hong et al., 2022b, Lemma 6). Analogously, for $\phi_k$ we can derive the following.

$$\sum_{k=1}^{K}\sum_{h=1}^{H}\sum_{j=1}^{d_Y}\mathbb{1}\{N_k(X_{k,h}[z_j]) > H\}\phi_k(X_{k,h}[z_j])$$

$$\le d_Y NH\sqrt{N^{d_X}KH\log(4Kd_Y N^Z)\log\left(1 + \frac{KH}{2N^{d_X}\Lambda_{0,z}}\right)}$$

Notice that again, due to the $Z$-sparseness, we can replace in the steps above $N^{d_X}$ with $N^Z$. Combining the terms, we have:

$$\sum_{k=1}^{K}\sum_{h=1}^{H}\sum_{j=1}^{d_Y}\beta_k(X_{k,h}[z_j]) \leq Hd_Y N^{d_X} + 2d_Y NH \sqrt{N^{d_X} KH \log\left(4Kd_Y N^Z\right) \log\left(1 + \frac{KH}{2N^{d_X}\Lambda_{0,z}}\right)}$$

$\square$

## F    ADDITIONAL EXPERIMENT WITH VARYING PRIOR KNOWLEDGE

In Section 6, we reported results for C-PSRL with a fixed degree of prior knowledge $\eta = 2$. It is interesting to see how changing the degree of prior knowledge affects the regret of C-PSRL. To this end, in Figure 3, we report an additional experiment in the same setting of Figures 2a, 2b where we compare C-PSRL with $\eta \in \{1, 2, 3, 4\}$ against F-PSRL, which corresponds to C-PSRL with $\eta = 5$, meaning the full graph is provided as an input to the algorithm. Note that we omit PSRL from the plot here to get a clearer view of the regret curves.

Perhaps surprisingly, varying the prior causal knowledge does not affect the regret of C-PSRL in a significant way. This may look discordant with the result in Theorem 4.1. However, we note that this domain is extremely small, so that the regret rate is actually dominated by the first term. Investigating the fine-grained impact of $\eta$ in the regret of C-PSRL in larger domains (especially when $d_X \gg Z$) would be a nice corroboration of our theoretical analysis, which we leave as future work.

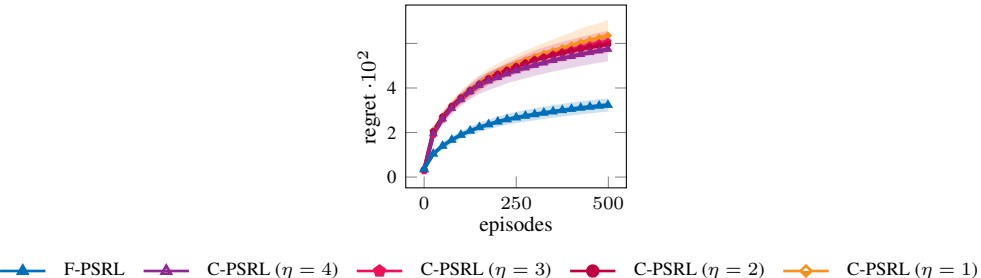

Figure 3: Regret and model error as a function of the episodes in the Random FMDP domain with $d_X = 9, d_Y = 6, Z = 5, N = 2, H = 100$. The plots report the mean and 95% c.i. over 20 runs.

## G  NOTE ON CONFOUNDING

In this paper, we assume there is no confounding acting on the variables of a causal graph $\mathcal{G} = (\mathcal{X}, \mathcal{Y}, z)$. While this assumption brings the proposed problem formulation closer to the literature on FMDPs (Osband & Van Roy, 2014; Xu & Tewari, 2020; Tian et al., 2020; Chen et al., 2020; Talebi et al., 2021; Rosenberg & Mansour, 2021), we believe that extending our analysis to include confounding would further narrow the gap between our formulation and real-world applications, which may admit confounding. Here, we report a few notes on how confounding affects our results and the additional challenges it brings, while we leave as future work a formal study on confounding and how to deal with it.

Let us waive the no confounding assumption and admit the presence of a set of *unobserved* random variables $\mathcal{C} = \{C_j\}_{j=1}^{d_C}$ taking values $c_j \in [N]$. In general, we can identify three types of confounding according to how the unobserved variables $\mathcal{C}$ interact with the observed variables $\mathcal{X}, \mathcal{Y}$:

(1) Confounding on $\mathcal{X}$, i.e., there are causal edges of the form $X_i \leftarrow C_j \rightarrow X_k$ for some $j \in [d_C]$ and $i, k \in [d_X]$. This can be equivalently modeled through bi-directed edges in $\mathcal{X} \times \mathcal{X}$;

(2) Confounding between $\mathcal{X}$ and $\mathcal{Y}$, i.e., there are causal edges of the form $X_i \leftarrow C_j \rightarrow Y_k$ for some $i \in [d_X]$, $j \in [d_C]$, and $k \in [d_Y]$. This can be equivalently modeled through bi-directed edges in $\mathcal{X} \times \mathcal{Y}$;

(3) Confounding on $\mathcal{Y}$, i.e., there are causal edges of the form $Y_i \leftarrow C_j \rightarrow Y_k$ for some $j \in [d_C]$ and $i, k \in [d_Y]$. This can be equivalently modeled through bi-directed edges in $\mathcal{Y} \times \mathcal{Y}$.

The type (1) can be easily incorporated in our analysis, as the confounding does not affect transitions in any meaningful way. Specifically, our regret result (Theorem 4.1) would stand without changes, whereas the causal discovery result (Corollary 5.1) would be slightly weakened. In particular, we could still identify the edges in $\mathcal{X} \times \mathcal{Y}$ since we observe all parents of each node in $\mathcal{Y}$, however, the edges in $\mathcal{X} \times \mathcal{X}$ cannot be discovered by our procedure.

The type (2) is arguably the most challenging: The confounding directly impact the transition probabilities as $P(\mathcal{X}, \mathcal{Y}) = p(\mathcal{Y}|\mathcal{X}, \mathcal{C})p(\mathcal{X}|\mathcal{C})p(\mathcal{C})$, where the conditioning $\mathcal{C}$ is unobserved. Our feeling is that this case brings the model close to a Partially Observable MDP (POMDP, Åström, 1965) formulation. Unfortunately, learning in POMDPs is known to be intractable in general (e.g., Krishnamurthy et al., 2016), which means further structural assumptions shall be considered to deal with this type of confounding.

The type (3) can also be problematic. With this type of confounding, we do not observe all of the causal parents of the variables in $\mathcal{Y}$, which may complicate modeling the transition probabilities in some (rare) cases.

