# OpenReview forum: "Exploiting Causal Graph Priors with Posterior Sampling for Reinforcement Learning"
_ICLR.cc/2024/Conference — ICLR 2024 poster_

### Official Review · Reviewer_i58j · 2023-10-29

**Soundness:** 3 good
**Presentation:** 2 fair
**Contribution:** 3 good
**Rating:** 8
**Confidence:** 3

**Summary:**

This work proposes a posterior sampling algorithm to address the reinforcement learning problem given a partial causal graph. The method utilizes the knowledge of a partial causal graph to construct a set of plausible factored MDPs (FMDPs). Then, it is framed as a hierarchical Bayesian model, with a prior over the factorization (equivalently, the complete causal graph) at a higher level and priors over the transitions and rewards at a lower level (given the factorization). The proposed posterior sampling C-PSRL algorithm proceeds by first sampling a factorization from the posterior, then sampling the MDP's parameters from the posterior given the factorization, and computing the optimal policy given the sampled FMDP. The paper proves that the Bayesian regret of the proposed C-PSRL algorithm is sublinear in the number of episodes and shows that C-PSRL can identify a super-graph of the true causal graph.

**Strengths:**

The motivation behind tackling the RL problem using a partial causal graph is compelling and essential for improving the practicality of posterior sampling algorithms in the RL context. The proposed algorithm is intuitive and incorporates Bayesian regret analysis.

Although the theoretical analysis is based on the PSRL approach in FMDPs in Osband & Van Roy (2014) and the hierarchical posterior sampling method in Hong et al (2022b), the paper analyzes the connection between the degree of prior knowledge in the partial causal graph and the Bayesian regret, which is interesting. The paper also provides a result on the weak causal discovery.

**Weaknesses:**

1. The experiment section is quite brief. There may be some missing details.

   a. In particular, the construction of priors is not discussed in detail. Although the priors are discussed in lines 173-178, it is not very clear to me about the distribution of the latent hypothesis space. For example, how do we select the support of this distribution (given the sparseness Z) and assign the prior probability of each factorization? What are the priors of the transition probabilities and rewards? How are the posterior distributions updated (do we need conjugate priors)?

   b. Is the assumption in Lemma D.1 satisfied by the baseline with uninformative priors in the experiments?

   c. Additionally, it might be beneficial to include an experiment involving F-PSRL with the causal graph set to the partial causal graph.

2. Could the author provide justification for the assumption made in Definition 2 that any misspecification in $\mathcal{G}_{\mathcal{F}_k}$ negatively affects the value function of $\pi_k$?

A minor issue: in line 180, line 9 does not exist in Algorithm 1.

**Questions:**

Please address the above weaknesses.

---

> ### Author Response · Authors · 2023-11-16
> **Authors Response**
>
> We thank the Reviewer for the positive feedback and for pointing out some missing details that we are incorporating in the updated version of the paper. We provide detailed comments below.
>
> **a. Priors and updates**
> *The construction of priors is not discussed in detail. Although the priors are discussed in lines 173-178, it is not very clear to me about the distribution of the latent hypothesis space. For example, how do we select the support of this distribution (given the sparseness Z) and assign the prior probability of each factorization? What are the priors of the transition probabilities and rewards? How are the posterior distributions updated (do we need conjugate priors)?*
>
> Given the limited space, we deferred the details on how to construct the priors and posterior updates to Appendix B. We better highlighted this in the text.
>
> In brief, the hyper-prior is defined through $d_Y$ categorical distributions, each one defining the probability of a specific causal parents assignment for a variable $Y_j$ in $\mathcal{Y}$. As common in previous works, the prior of the transition probabilities are modeled through Dirichlet distributions (conditioned on the hyper-prior) for each $\mathcal{X}$ assignment. For the ease of presentation, the reward function is assumed to be known in the paper, and thus we do not specify a prior. Typically, the latter is specified through a Beta distribution in previous works (and in our regret analysis in the appendix). Since we are using conjugate priors, we can compute the posterior updates in closed form, as detailed in Appendix B.
>
> **b. Is the assumption in Lemma D.1 satisfied by the baseline with uninformative priors in the experiments?**
>
> Yes, the baseline with uninformative prior is a vanilla version of PSRL for tabular MDPs. The prior over the transition parameters are still specified through a Dirichlet distribution for every state-action assignment.
>
> **c. Additionally, it might be beneficial to include an experiment involving F-PSRL with the causal graph set to the partial causal graph.**
>
> This is an interesting point! Actually, this experiment is related to prior misspecification, a setting we did not cover in the paper. Crucially, the regret of F-PSRL is not guaranteed to be sub-linear under prior misspecifications. Anyway, for the sake of completeness, we are going to run this experiment, which we will add to the appendix in a final version of the paper.
>
> **2. Could the author provide justification for the assumption made in Definition 2 that any misspecification in $\mathcal{G}_{\mathcal{F}_k}$ negatively affects the value function of $\pi_k$?**
>
> Yes, this assumption basically means that the causal edges in the true causal graph are *necessary* and missing one of them will not result in the optimal policy being recovered from the algorithm. This is not always true: The optimal policy may traverse a tiny portion of the FMDP, making some of the variables irrelevant. In the latter case, we cannot hope to learn the causal edges of the irrelevant variables. Instead, those variables shall just be omitted from the learning process.
>
> **A minor issue: in line 180, line 9 does not exist in Algorithm 1.**
>
> Great catch, we have corrected this. Thanks.

---

> > ### Comment · Reviewer_i58j · 2023-11-20
> >
> > Thank you for the detailed reply. I am satisfied with both the response and the commitment of the authors.

---

### Official Review · Reviewer_mWkL · 2023-11-01

**Soundness:** 3 good
**Presentation:** 3 good
**Contribution:** 3 good
**Rating:** 8
**Confidence:** 3

**Summary:**

This work proposes an extension of Osband & Van Roy (2014, NeurIPS) where only partial knowledge of the causal graph is needed as opposed to full knowledge. More specifically, the aim is to still provide competitive rates in terms of Bayesian regret with such partial knowledge. The authors also find that their method also performs a weak notion of causal discovery as a byproduct.

**Strengths:**

- An extension of an interesting frameowork
- clear statement about what are the new contributions

**Weaknesses:**

Some issues in clarity of presentation and about what are the interventions here ? The paper is not written in causality notation
- Not clear how much computational complexity is added wrt to PSRL; eg what is the complexity of step 2 in Algo 1 ?
- defintions in Sec 2.1 are currently wrong or incomplete: the Markov factorization is not what makes an CBN, since it applies to standard BNs. See definitions of CBN in eg [Pearl, 2009, Definition 1.3.1] , it's about *all* interventional distributions having the Markov fact. and other constraints.
- I'm not understanding in Alg 1 about step 7, are these posteriors in closed form ? If not, what approximations are used and how do they affect the regret ?

**Questions:**

- Could you compare with the regret bounds of Lattimore et al (2016) that also use knowledge of the causal structure to improve regret in the context of bandits instead of RL ? Are there connections?  See also the extension with unknown graph in Lu et al (2021)



- Lattimore, F., Lattimore, T. & Reid, M. D. Causal bandits: learning good interventions via causal inference. In Adv. Neural Information Processing Systems Vol. 29 (2016)
- Lu, Y., Meisami, A. and Tewari, A., 2021. Causal bandits with unknown graph structure. Advances in Neural Information Processing Systems, 34, pp.24817-24828.

---

> ### Author Response · Authors · 2023-11-16
> **Authors Response**
>
> We want to thank the Reviewer for the useful feedback, which allows us to improve clarity of the presentation and to include some additional related references. We provide detailed replies below.
>
> **Computational complexity.**
> *Not clear how much computational complexity is added wrt to PSRL; eg what is the complexity of step 2 in Algo 1?*
>
> This is a great point! Building the set of consistent factorizations requires a significant burn-in computational cost. To be specific, our method allows to build the set of consistent parents for each variable $Y_j$ independently, which means the process can be parallelized on $d_Y$ workers. For each worker, we need to build a set of $\sum_{i = 0}^{Z - \eta} \binom{d_X - \eta}{i}$ elements, which we can do recursively by calling the base function at most $O(2^{d_X - \eta})$ times.  Notably, this is not the computational bottleneck of the algorithm, but the planning procedure (Algorithm 1, line 6) to get $\pi_k$, which is executed for every episode. Thus, the computational complexity of C-PSRL is fundamentally the same of standard PSRL for factored MDPs in terms of rate. How to avoid the computational burn-in (e.g., not pre-computing the whole set of factorizations) would be a nice direction for future works.
>
> **Definition of causal graphical mode.l**
> *Definitions in Sec 2.1 are currently wrong or incomplete: the Markov factorization is not what makes an CBN, since it applies to standard BNs. See definitions of CBN in eg [Pearl, 2009, Definition 1.3.1], it's about all interventional distributions having the Markov fact. and other constraints.*
>
> We agree that the current definition is somewhat too informal. Although the constraint on all the interventional distributions is kind of natural in the FMDP formulation, we updated the text to make it explicit and to avoid any confusion. Thanks for pointing this out!
>
> **Posterior updates.**
> *I'm not understanding in Alg 1 about step 7, are these posteriors in closed form? If not, what approximations are used and how do they affect the regret?*
>
> Yes, for our choice of priors, those posteriors are computed in closed form. While the priors definition and posterior updates are detailed in Appendix B, we updated the text to highlight this in the main paper as well.
>
> **Causal bandits.**
> *Could you compare with the regret bounds of Lattimore et al (2016) that also use knowledge of the causal structure to improve regret in the context of bandits instead of RL ? Are there connections? See also the extension with unknown graph in Lu et al (2021)*
>
> Whereas both our setting and causal bandits consider causal graphical models in the context of sequential decision making, there are some crucial differences. Causal bandits assume the presence of a causal graphs over a set of variables. Within these variables, there is one reward generating variable $X_R$ and the goal of the algorithm is to identify $X_R$ through interventions and to minimize the worst-case simple regret. In our setting: (1) we assume the presence of a causal structure on the transition dynamics rather than the reward, (2) we consider a Bayesian setting in which the causal structure is partially known, (3) we evaluate our algorithm on the Bayesian $K$-episode regret instead of the frequentist simple regret. These aspects make our results incomparable to causal bandits literature. Anyway, we are happy to add a discussion on differences and similarities in the updated related work section.

---

> > ### Comment · Reviewer_mWkL · 2023-11-16
> >
> > Thank you for the thorough response and for updating a revised paper.
> > I am happy with the authors' response therefore I increase my score.

---

### Official Review · Reviewer_fSwc · 2023-11-02

**Soundness:** 3 good
**Presentation:** 3 good
**Contribution:** 2 fair
**Rating:** 6
**Confidence:** 2

**Summary:**

The paper proposes a novel approach to posterior sampling for reinforcement learning (PSRL), namely C-PSRL, that exploits the partial causal graph (which is given as prior knowledge) as a prior. C-PSRL iteratively samples the causal graph which induces FMDP and updates the posterior. As a result, C-PSRL improves the Bayesian regret to $\tilde{\mathcal{O}}(\sqrt{K/2^\eta})$ where $\eta$ represents a degree of prior knowledge. An interesting property of C-PSRL is a weak causal discovery, i.e., it discovers the sparse super-graph of the ground-truth causal graph as a byproduct. Experimental results demonstrate the effectiveness of the proposed algorithm.

**Strengths:**

- The manuscript is well-written. As one who is not an expert in PSRL literature, it was not too difficult to follow the paper.
- The problem setting where only the partial causal graph is given is interesting. Also, the motivation is clear; in many practical scenarios complete true causal graph is often not accessible but only partial causal relationships are known.
- The paper suggests a neat solution that extends PSRL and provides the Bayesian regret analysis, though I did not check the proofs. Empirical validation demonstrates its effectiveness compared to naive PSRL.

**Weaknesses:**

- It is unclear what is the implications of the weak causal discovery result. It states that the $Z$-sparse super-graph of the ground-truth causal graph can be recovered, but it does not speak for itself, and difficult to understand what is the takeaway message. For example, is it fundamentally impossible for the C-PSRL to recover the ground-truth causal graph? If this is the case, then is it still possible for the C-PSRL to obtain (converge to) optimal policy? Is the degree of the difference between the true causal graph and super-graph associated with the convergence rate or optimality?
- The assumption that the (correct) partial causal graph is given as a priori is understandable. However, it is also important to discuss what would happen if this assumption does not hold. For example, how would C-PSRL behave if the initial partial causal graph is wrong (i.e., when some of its edges are non-causal)?

**Questions:**

- How can the byproduct (i.e., super-graph of the ground-truth causal graph) be utilized? Also, if one is interested in recovering the ground-truth causal graph, do you think there is any way to further prune the super-graph to obtain the true graph?
- It would be interesting to see the performance of C-PSRL given the low prior knowledge (e.g., $\eta = 0, 1$). The experiment only shows the case of $\eta=2$.

---

> ### Author Response · Authors · 2023-11-16
> **Authors Response**
>
> We thank the Reviewer for the thoughtful comments on the weak causal discovery result and possible prior misspecifications. We provide detailed answers below.
>
> **Weak causal discovery.**
> *Is it fundamentally impossible for the C-PSRL to recover the ground-truth causal graph? If this is the case, then is it still possible for the C-PSRL to obtain (converge to) optimal policy? Is the degree of the difference between the true causal graph and super-graph associated with the convergence rate or optimality?*
>
> This is an important aspect: C-PSRL does not have any incentive to prune non-causal edges from the recovered graph as long as they do not hurt the performance of the resulting policy, which means that the true causal graph cannot be recovered in general.
>
> C-PSRL can definitely still converge to the optimal policy. Indeed, the set of FMDPs consistent with a super-graph include the set of FMDPs consistent with the true graph. Thus, the algorithm can still explain the transition model well starting from the super-graph, and then recovering the optimal policy from the transition model.
>
> In our analysis, the regret rate is totally unrelated to the difference between the true causal graph and the super graph.
>
> **Wrong edges in the prior.**
> *The assumption that the (correct) partial causal graph is given as a priori is understandable. However, it is also important to discuss what would happen if this assumption does not hold. For example, how would C-PSRL behave if the initial partial causal graph is wrong (i.e., when some of its edges are non-causal)?*
>
> This is a very interesting topic. As it is typical in Bayesian methods, C-PSRL assumes that the prior is correct, which means nothing changes in the algorithm if there are misspecifications in the initial graph.
> However, our regret guarantees do not hold in the latter setting. Studying how prior misspecifications impacts the regret is a nice direction for future works (e.g., see Simchowitz et al., Bayesian decision-making under misspecified priors with applications to meta-learning, 2021).
>
> In practice, one shall include in the initial prior only the causal edges that are reliable, leaving to the algorithm the task of discovering the other important edges.
>
> **How to use the super-graph.**
> *How can the byproduct (i.e., super-graph of the ground-truth causal graph) be utilized?*
>
> For instance, one could use the obtained super-graph for explainability, e.g., what are the factors that make the policy take a specific decision in a given circumstance. Another possible use of the super-graph is to warm-start a subsequent causal discovery procedure to recover the true casual graph (see below).
>
> *Also, if one is interested in recovering the ground-truth causal graph, do you think there is any way to further prune the super-graph to obtain the true graph?*
>
> Whereas the true graph cannot be recovered with C-PSRL as currently designed, we believe there are a few potential options if one is interested in obtaining the true graph:
> - One could add in the C-PSRL objective a specific penalization for the number of edges in the considered factorization, fostering the algorithm to avoid unnecessary edges. It is not immediately clear how this would affect the regret rate and the weak causal discovery result. This is an interesting matter for future works;
> - One could run a causal discovery procedure warm-started with the super-graph obtained through C-PSRL, e.g., running conditional independence tests to further prune the graph. Note that running those tests is not trivial, as the problem is inherently sequential and one cannot just set the state variables in $\mathcal{X}$, but has to deploy a policy that drives the MDP to that desired state.
>
> **It would be interesting to see the performance of C-PSRL given the low prior knowledge (e.g., = 0, 1). The experiment only shows the case of = 2.**
>
> We will include these experiments in the appendix of a final version of this paper. From our experience, small variation of $\eta$ is unlikely to make a significant difference in low-dimensional domains as those presented in the paper (e.g., 6 binary variables for the states and 3 binary variables for the actions), in which the hyper-prior converges faster than the prior and does not critically affect the regret. Instead, it would be of great interest to show that varying $\eta$ can dramatically affect the regret in larger domains, as we expect. However, tackling those kinds of domains would require a substantial effort to design tractable approximations of the transition model, posterior updates, and planning procedures. We believe the latter fall beyond the scope of this paper and it is a nice matter for future works.

---

> > ### Comment · Reviewer_fSwc · 2023-11-20
> >
> > Thanks for the detailed feedback. I have no further questions so far.

---

### Official Review · Reviewer_m9sB · 2023-11-02

**Soundness:** 3 good
**Presentation:** 3 good
**Contribution:** 3 good
**Rating:** 8
**Confidence:** 4

**Summary:**

Posterior sampling guarantees sample efficiency by exploiting prior knowledge of the environment's dynamics. In practice, the class of parametric distributions can be cumbersome and uninformative. This paper proposes a hierarchical Bayesian procedure, C-PSRL, that uses the (partial) causal graph over the environment's variables as the prior. Its Bayesian regret is associated with the degree of prior knowledge. Empirical experiments show that C-PSRL improves the efficiency of posterior sampling with an uninformative prior.

**Strengths:**

The idea of replacing an uninformative prior with a causal graph is attractive. Additionally, the byproduct of weak causal discovery reveals the possibility of how to utilize reinforcement learning in the causal discovery field.

**Weaknesses:**

My biggest concern is why not learn directly from the parents of the reward, given that we want to utilize causal prior knowledge? If I understand correctly, this paper focuses more on the parents of actions and states. As we all know, the value of a variable depends only on the values of its causal parents (line 122). If our goal is to learn the policy that maximizes the reward, shouldn't we primarily focus on the parents of the reward for greater efficiency and simplicity?

**Questions:**

1. Line 92: I'm a little confused about why the DAG is divided into X and Y parts. If X is the state-action space and Y is the state space, where is the reward in the DAG? In common DAGs, each node represents a random variable, but here, each node seems more like a realization of the tuple (s, a) or just state 's', which makes me doubt the interpretability of the DAG.
2. Figure 1 (right): The set Z includes 9 different DAGs. However, shouldn't the set of factorizations consistent with G_0 contain 2x2x3=12 possible FMDPs? Or did I miss some selection criteria for the set Z?
3. Algorithm 1: From line 5 to line 6, could you provide more details on how to obtain F_* from F_k?

---

> ### Author Response · Authors · 2023-11-16
> **Authors Response**
>
> We want to thank the Reviewer for the positive feedback and thoughtful questions that allow us to clarify some important aspects of the paper.
>
> **Why not learn directly from the parents of the reward, given that we want to utilize causal prior knowledge?**
>
> This is a great question! Indeed, our approach is *model-based*, which is by far the most common way posterior sampling has been employed in RL. In a model-based paradigm, we aim to learn a model of the transition dynamics from which the optimal policy can be extracted, instead of learning the policy (or the value function) directly. Thus, only the causal parents of the transition model matter.
>
> However, a few instances of *model-free* implementations of posterior sampling for RL also exist (Dann et al., A Provably Efficient Model-Free Posterior Sampling Method for Episodic Reinforcement Learning, 2021; Tiapkin et al., Model-free Posterior Sampling via Learning Rate Randomization, 2023). In a model-free paradigm, we directly learn the optimal policy or value function from data. Extending our approach to handle (partial) causal priors over the reward or value function in a model-free setting looks like a nice direction for future work, which we are happy to mention in the updated final section of the paper.
>
> Nonetheless, we want to point out two crucial aspects for which a model-based approach can be preferred. First, to the best of our knowledge, the optimal rate of model-based PSRL has not been matched by model-free versions. Model-based PSRL is actually believed to be minimax optimal in tabular settings (Osband \& Van Roy, 2017). Secondly, in a model-based paradigm, the same prior knowledge over the transition dynamics can be exploited to solve multiple tasks (with different reward functions). For instance, having some causal knowledge over the model of patients in clinical trials, we may try to learn optimal treatment strategies of different diseases (i.e., rewards).
>
> **Question 1.**
> *Line 92: I'm a little confused about why the DAG is divided into X and Y parts. If X is the state-action space and Y is the state space, where is the reward in the DAG? In common DAGs, each node represents a random variable, but here, each node seems more like a realization of the tuple (s, a) or just state 's', which makes me doubt the interpretability of the DAG.*
>
> In our representation, each node of the DAG is a random variable. The variables in $\mathcal{X}$ represent state and action features at the time step $t$, while the variables in $\mathcal{Y}$ represent the state features at the step $t +1$. This is the standard representation in FMDPs literature and, in our opinion, the most natural way to express the transition model through a DAG.
>
> In the paper, we omit modelling the reward for the ease of presentation. In presenting the algorithm and the experiments, we consider the case in which the reward function is known. In the regret analysis, we consider an unknown reward $R(X)$ with the same causal parents of the transition model. Extending the algorithm and the analysis to handle unknown rewards with specific causal parents is straightforward, and does not have meaningful impact on the presented results as long as other assumptions hold (sparseness, degree of prior knowledge, finite support).
>
> To include the reward in the DAG, one can introduce an additionl variable in $\mathcal{Y}$ (representing the reward) with causal parents in $\mathcal{X}$.
>
> **Question 2.**
> *Figure 1 (right): The set Z includes 9 different DAGs. However, shouldn't the set of factorizations consistent with $G_0$ contain 2x2x3=12 possible FMDPs? Or did I miss some selection criteria for the set Z?*
>
> In the reported example, the set of factorizations consistent with $\mathcal{G}_0$ are the $3$-sparse DAGs that include all of the edges in $\mathcal{G}_0$ (and possibly some additional edges).
>
> **Question 3.**
> *Algorithm 1: From line 5 to line 6, could you provide more details on how to obtain $F_\star$ from $F_k$?*
>
> Let us clarify that $F_\star$ is unknown and cannot be obtained from $F_k$. However, we can collect samples from the true FMDP, as it is typical in RL. The Line 6 of the pseudocode means that an episode is collected by deploying the policy on the true FMDP $F_\star$ without knowing it.

---

> > ### Comment · Reviewer_m9sB · 2023-11-21
> >
> > Thank you for the detailed response. I have no further questions.

---

### Author Response · Authors · 2023-11-16
**Global Response**

We want to thank the Reviewers for their valuable feedback, which we incorporated in the updated version of the paper to further improve the presentation. We are more than happy to hear that they found the motivation to be "clear" (Reviewer fSwc) and "compelling" (Reviewer i58j), our framework to be "interesting" (Reviewers fSwc, mWkL, i58j) and "attractive" (Reviewer m9sB), our approach a "neat solution" (Reviewer fSwc) to the problem, which is "essential for improving the practicality of posterior sampling algorithms in the RL context" (Reviewer i58j).

We hope that our replies will make them appreciate our paper even more, and we will be happy to take any further question they might have during the discussion period.

**What is new in the updated version**

We uploaded a new version of the paper following Reviewers' suggestions to improve the presentation. All of the changes are highlighted in blue. In brief:
- We have included a clarification on the definition of graphical causal model in lines 98-103, which better aligns our definition with the literature;
- While describing the pseudocode of C-PSRL (lines 193-196), we better highlighted that the posterior updates can be computed in closed form, as detailed in Appendix B;
- In the related work section, we mentioned the literature on causal bandits (lines 351-359), which also consider a causal structure in sequential decision-making, but it is otherwise orthogonal to our work;
- We extended the conclusions (lines 396-399) to include some of the ideas for future directions emerged from the reviews.

---

> ### Author Response · Authors · 2023-11-22
>
> With the discussion period approaching its end, we want to thank the Reviewers for acknowledging our response and for their appreciation of the paper.
>
> Kind regards,
>
> The Authors

---

### Meta-Review · Area_Chair_dnkD · 2023-12-22

**Metareview:**

This work considers an extension of PSRL in which one has a prior on the causal graph of the transition dynamics, and provide Bayesian regret bounds for this setup. The idea is straightforward, the exposition is clear, and reviewers found the connection between RL and causal discovery to be interesting. I encourage the authors to discuss topics brought up by the reviewers in the CR, as this can help with providing useful background and contextualize some of the assumptions made in this work.

I personally enjoyed this work, but feel like it could benefit from a few more experiments, perhaps with a setup inspired by some of Susan Murphy's work used to motivate the paper in the introduction.

**Justification For Why Not Higher Score:**

The paper is rather straightforward and the experiments are somewhat limited.

**Justification For Why Not Lower Score:**

Reviewers liked it and it makes a nice connection between RL and causal discovery.

---

### Decision · Program_Chairs · 2024-01-16

Accept (poster)